# VITRIX-UniViTAR: Unified Vision Transformer with Native Resolution

**Limeng Qiao    Yiyang Gan    Bairui Wang    Jie Qin    Shuang Xu    Siqi Yang    Lin Ma**[⊠]

*Meituan Inc.*

qiaolm@pku.edu.cn, realgump@tju.edu.cn, {tjbairuiwang, jayqinliu}@gmail.com
sxu1997@126.com, siqi.yang@uq.net.au, forest.linma@gmail.com

## Abstract

Conventional Vision Transformer streamlines visual modeling by employing a uniform input resolution, which underestimates the inherent variability of natural visual data and incurs a cost in spatial-contextual fidelity. While preliminary explorations have superficially investigated native resolution modeling, existing works still lack systematic training recipe from the visual representation perspective. To bridge this gap, we introduce **Uni**fied **Vi**sion **T**ransformer with N**A**tive **R**esolution, *i.e.* UniViTAR, a family of homogeneous vision foundation models tailored for unified visual modality and native resolution scenario in the era of multimodal. Our framework first conducts architectural upgrades to the vanilla paradigm by integrating multiple advanced components. Building upon these improvements, a progressive training paradigm is introduced, which strategically combines two core mechanisms: *(1)* resolution curriculum learning, transitioning from fixed-resolution pretraining to native resolution tuning, thereby leveraging ViT's inherent adaptability to variable-length sequences, and *(2)* visual modality adaptation via inter-batch image-video switching, which balances computational efficiency with enhanced temporal reasoning. In parallel, a hybrid training framework further synergizes sigmoid-based contrastive loss with feature distillation from a frozen teacher model, thereby accelerating early-stage convergence. Finally, trained exclusively on public accessible image-caption data, our UniViTAR family across multiple model scales from 0.3B to 1.4B achieves state-of-the-art performance on a wide variety of visual-related tasks. The code and models are available here.

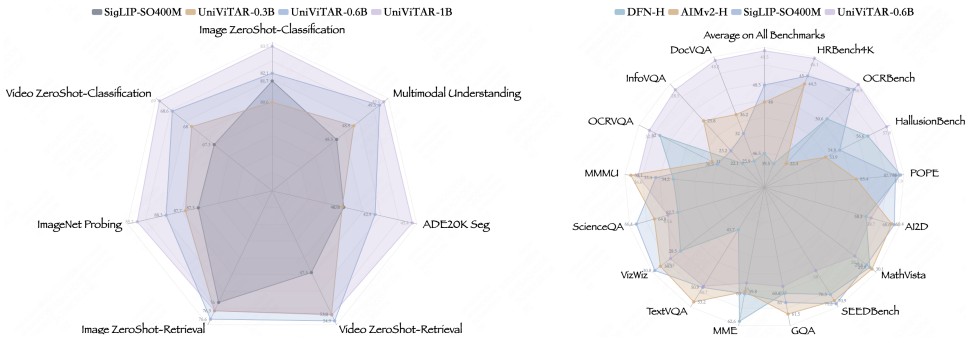

Figure 1: The figure presents: (left) a systematic overview of model scaling performance across downstream tasks when increasing parameter size from 0.3B to 1B, and (right) a comprehensive comparison of multimodal capabilities against SOTA baselines on diversified benchmarks.

39th Conference on Neural Information Processing Systems (NeurIPS 2025).

# 1 Introduction

In the era of rapid advancements of multimodal large models, Vision Transformer [1], characterized by its simplicity and scalability, has emerged as a foundational architecture for visual representation learning. Drawing inspiration from transformer-based large language models, conventional ViT usually uniformly converts raw visual data into square aspect ratio and standardized resolution to reduce modeling complexity and simplify the training workflow. While this paradigm simplifies feature extraction and aligns with existing engineering practices, it inherently imposes artificial constraints on real-world visual data by disregarding the inherent variability of natural images.

Recent studies have preliminarily investigated the vision backbone within a native resolution paradigm. FlexViT [2] introduces a flexible ViT architecture featuring dynamical patch size selection in the patch embedding layer, which facilitates smooth variation of token sequence length through parametric scaling. In contrast, NaViT [3] maintains fixed patch size while directly processing native resolution images with varying aspect ratios, where the token sequence length of different images changes dynamically. This approach demonstrates the feasibility and benefits of adopting natural language processing style packing strategies for vision foundational model. Qwen-VL's [4, 5] vision encoder inherits NaViT's core configuration while specifically investigating native resolution impacts from a multimodal large model perspective. While the aforementioned approaches have attracted initial research attention, the field still lacks a comprehensive series of architecture-homologous vision backbones that can simultaneously support native- and fixed-resolution processing, achieve high-fidelity feature extraction for both images and videos.

To address this gap, we present the **Uni**fied **Vi**sion **T**ransformer with N**A**tive **R**esolution, termed as **UniViTAR**, a family of vision foundational backbones designed to uniformly process visual modalities (image or video) with native resolution and dynamic aspect ratio. Building upon insights from large language model recent practices and architectural innovations in visual transformers, our approach firstly conduct systematic architectural upgrades to the vanilla ViT paradigm by integrating multiple advanced components: 2D Rotary Position Embedding, SwiGLU activation function, RMSNorm layer, QK-Norm mechanism, and LayerScale module. These modifications collectively establish a more robust architectural foundation compared to conventional implementations. Secondly, we develop a progressive training paradigm with two complementary adaptation strategies: *1)* the progressive resolution adaptation strategy employs curriculum learning from fixed low-resolution (e.g., 224) pretraining to native-resolution fine-tuning. Notably, our experiments reveal that the advanced ViT architecture exhibit remarkable adaptability - models pretrained at fixed resolution can efficiently generalize to variable-length visual sequences through limited native resolution tuning. *2)* the progressive visual modality adaptation strategy addresses computational challenges in video processing by deferring video data integration to the final training phase. We further demonstrate that alternating image-video training sequences (inter-batch modality switching) significantly outperforms mixed-batch (intra-batch modality mixing) in preserving image understanding capabilities while acquiring temporal reasoning skills. Thirdly, we implement a hybrid training framework combining contrastive learning objectives with distillation techniques. Our primary optimization employs a sigmoid-based contrastive loss [6] for unified image-video representation learning. To accelerate early-stage convergence, we further incorporate feature distillation from a frozen vision teacher model as an auxiliary training objective during initial phases, then gradually phasing out this regularization as the model matures. Finally, through this comprehensive approach trained on public-accessible datasets, we successfully scale a family of vision backbones supporting native resolutions and both visual modalities, with parameter counts ranging from 0.3B to 1.4B. Extensive evaluations demonstrate the effectiveness of our proposed methods.

Specifically, the contributions of our UniViTAR family are summarized as follows:

- We introduce a family of homogeneous visual foundation models that support native resolution and unified feature extraction across visual modalities, offering the community a versatile framework for multimodal research.

- We develop an efficient and effective progressive training strategy that addresses the computational challenges of native resolution modeling while systematically enhancing the model's image-caption alignment capability.

- We train our models with public-accessible datasets, achieve leading performance with limited resources, and observe a trend of performance increasing with parameter scaling.

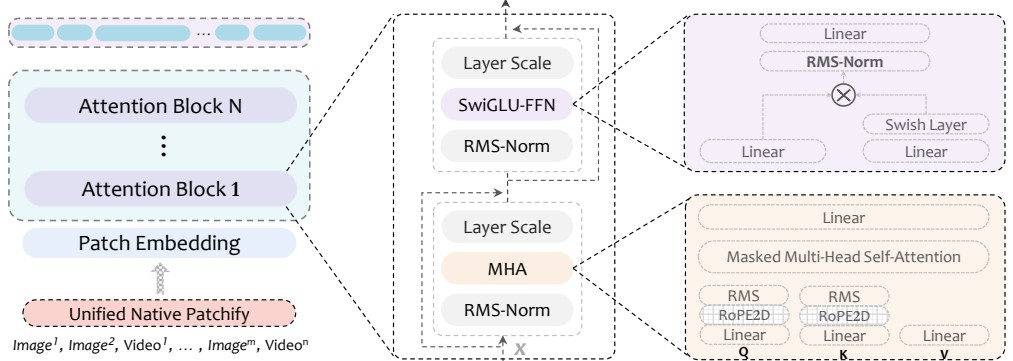

Figure 2: **Architecture of UniViTAR family.** All visual inputs are uniformly transformed into patch sequences and fed into Vision Transformer. In addition to using the *Pre-Norm* approach, we also adopt *RMS-Norm* as the normalization layer in both *MHA* and *FFN* module.

## 2 Method

### 2.1 UniViTAR: Homologous Visual Foundation Model

#### 2.1.1 Architecture Design

UniViTAR is a Transformer-based encoder model that inherits the original architecture of the conventional Vision Transformer [1] but incorporates the following advanced modifications:

***Unified Patchify for Native Image and Video Modality.*** As illustrated in the Figure 2, given the native input visual data $\mathbf{X} \in \mathbb{R}^{T \times C \times H \times W}$ of any vision modality (image, video), where $T = 1$ represents image and $T > 1$ represents video, **UniViTAR** firstly patchifies $\mathbf{X}$ into a series of dynamic length visual patch sequences $\mathbf{P} = (N, S)$, where $N$ is the number of patches per image/video and $S$ is the number of pixels per patch. Then a $3D$ convolution layer is adopted as the *Patch Embedding Layer* to consistently convert the above patch sequence into a visual token sequence $\mathbf{T} = (N, D)$, where $D$ is the hidden size of the following attention layers.

***2D Rotary Position Embedding.*** Drawing on the architecture designs of language models, the original ViT regards the position information among different visual tokens as a one-dimensional association. In fact, considering that visual data usually has spatial association (row and column) and temporal association (time), the position information between different tokens is usually considered to be multi-dimensional. Thence we remove the original absolute position encoding and introduce $2D$-RoPE [7] into each subsequent encoder layer to capture the two-dimensional positional information of images. Furthermore, we found that the presence or absence of the class token in the original ViT has almost no effect on model performance. To ensure the consistency of position encoding, we also empirically remove the design of class token.

***SwiGLU and RMSNorm.*** By leveraging the recent advances of LLaMA [8] architecture design for language modeling, UniViTAR incorporates SwiGLU as the feed-forward network (FFN) and replaces all normalization layers with RMSNorm. In addition, we adds an extra RMSNorm to each SwiGLU-FFN for good expressivity and improving the training stability.

***Query-Key Normalization.*** In order to improve the stability of model training, we adopt the QK-Norm technique [9, 10] , which applies normalization to the queries and keys before the dot-product attention computation, to directly controls the norm growth of input to the softmax and avoid abnormal attention logits. Note that we still utilize RMSNorm as the norm function to ensure the consistency of the architecture.

#### 2.1.2 Homologous Model Scaling

The UniViTAR family consists of a comprehensive suite of foundational and scratch-train models, encompassing a parameter range from 0.3 to 1.4 billion, *i.e.* UniViTAR-0.3B/0.6B/1B. The hyperparameters and important information are listed in Table 1 in details.

Table 1: **Detailed architectural configuration for UniViTAR family.**

| Model | Hidden-Size | Intermediate-Size | Encoder-Layers | Attention-Heads | Parameters (M) |
|---|---|---|---|---|---|
| UniViTAR-0.3B | 1024 | 4224 | 24 | 16 | **310** |
| UniViTAR-0.6B | 1280 | 5184 | 32 | 16 | **637** |
| UniViTAR-1B | 1920 | 7680 | 32 | 24 | **1419** |

Table 2: **Detailed training strategy illustration of UniViTAR family.**

| | Stage 1 | Stage 2 | Stage 3 | Stage 4 |
|---|---|---|---|---|
| Train Strategy | LLaMA · Contrastive · UniViTAR · Distill Loss · DinoV2 | LLaMA · Contrastive · UniViTAR | LLaMA · Contrastive · UniViTAR | LLaMA · Contrastive · UniViTAR |
| Data Modality | Image | Image | Image | Image, **Video** |
| Resolution | $224 \times 224$ | $224 \times 224$ | **Native** | **Native** |
| Loss Function | Sigmoid, KL | Sigmoid | Sigmoid | Sigmoid |
| Seen Samples | 12B | 1B | 1B | 0.6B |

## 2.2 Contrastive Vision-Language Pretrain with UniViTAR

### 2.2.1 Architecture Design

In general, the acquisition of UniViTAR largely follows the basic training paradigm of CLIP [11]. Specifically, the native-resolution visual input $v$ is encoded into the visual feature space via the UniViTAR encoder to obtain $F_v \in \mathbb{R}^{N_v \times D_v}$, while the textual input $t$ is projected into the textual feature space through a pretrained LLaMA [12] decoder to obtain $F_t \in \mathbb{R}^{N_t \times D_t}$. The dynamic-length visual features $F_v$ are then uniformly converted into the visual embedding $f_v \in \mathbb{R}^{D_v}$ through a global average pooling and the feature corresponding to the *<EOS>* token in $F_t$ is utilized as the textual representation $f_t \in \mathbb{R}^{D_t}$ of the input caption. Subsequently, $f_v$ and $f_t$ are further projected into the same shared semantic space via a linear projection layer respectively. Then a simple pairwise sigmoid loss [6] is employed as the contrastive supervision to align the visual and text modalities semantically.

### 2.2.2 Optimized Contrastive Training Strategy

To ensure that the model can converge efficiently and the training cost is controllable, we carefully design the training pipeline of UniViTAR into four stages in sequence, as shown in Table 2.

***Stage 1: Visual knowledge pre-acquisition with hybrid paradigm training.*** The primary objective of this phase is to efficiently pretrain a visual foundation model from scratch by integrating two classic learning paradigms: vision-text contrastive learning and visual knowledge distillation. Specifically, the proposed architecture employs a triple-branch parallel design: (1) a *xavier*-initialized UniViTAR, (2) a frozen pre-trained text encoder, and (3) a frozen pre-trained visual teacher. During training, only the target visual foundation model receives gradient updates, with other branches fixed to minimize computational overhead while preserving knowledge integration. For implementation, we adopt LLaMA [12] and DINOv2-g [13] as default components, though the framework supports substitution with alternative pre-trained foundation models. The composite training objective is defined as:

$$\mathcal{L}_{overall} = \mathcal{L}_{contrastive}(f_v^{UniViTAR}, f_t^{LLaMA}) + \lambda \cdot \mathcal{L}_{distillation}(f_v^{UniViTAR}, f_v^{Dino}) \quad (1)$$

where $\mathcal{L}_{contrastive}$ is the sigmoid loss from SigLIP [6] and $\mathcal{L}_{distillation}$ is the KL Divergence [14]. The target visual foundation model functions as a visual knowledge bridge, simultaneously performing image-text alignment and feature distillation. This phase processes 12B samples with all images resized to 224, constituting 82.2% of the total training data (12B/14.6B).

***Stage 2: Finetune with full-parameter for superior alignment.*** The objective of this stage is to further enhance the upper limit of image-text alignment through full-parameter fine-tuning of both vision and text encoders, establishing a unified semantic-visual space. The visual distillation branch is deactivated during optimization. Training employs identical image-caption pairs as Stage 1 at 224 resolution. Considering the high computational cost of full parameter fine-tuning, the training process is conducted on 1B samples, accounting for 6.9% of the total training data.

***Stage 3: Unlock the model-capacity of native-resolution.*** In this stage, our strategy extends the model capability to handle native-resolution, thereby achieving robust image-text alignment for dynamic-resolution inputs. However, enabling native-resolution capacity necessitates addressing two critical challenges: (1) ensuring positional encoding are thoroughly trained across variable sequence lengths, and (2) transfering feature distribution from uniformly-resized patches to native patches through efficient training. In practice, visual data is batched in its native form to preserve original resolutions and aspect ratios. Then the intra-batch images are dynamically scaled (with aspect ratios maintained) to align total sequence lengths $L_{total}$ with a predefined token limit $L_{max}$. That is to say, when the value of $L_{max}/L_{total}$ is greater than 1, all data will be uniformly enlarged, and vice versa the shape of all data will be reduced, ensuring consistent computational loads across batches. Within attention blocks, each token's receptive field is confined to tokens from the same image via masking, enabling isolated intra-image contextual modeling while preserving inter-sample independence. During training, resolution diversity within batches ensures comprehensive training of positional encoding across varying context lengths, progressively refining the model's ability to generate features aligned with native patch distributions. At inference, inputs are processed directly at their native resolutions without resizing. In this stage, 1B samples (6.85% of the total training) were trained with the text branch frozen throughout the process.

***Stage 4: Unifying visual modalities with image-video alternation training.*** The goal of this stage is to unify image and video input modalities with native-resolution and dynamic video length. Inspired by the InternVideo series [15, 16], we utilize both image-text and video-text pairs to optimize the UniViTAR checkpoint from Stage 3 with an image-video alternating training strategy. This strategy addresses three critical considerations: (1) leveraging image data's superior scale and diversity compensates for video data scarcity while maintaining visual content continuity; (2) joint image-video training preserves cross-modal comprehension capabilities; (3) alternating modality-specific updates enforce focused parameter adaptation through sequential modality optimization. The alternating training protocol first initiates each epoch with random permutation of image-video data to enhance stochasticity. Subsequently, data batches are partitioned into global batch units and alternately sequenced at global batch granularity. Through this structured approach, the configuration effectively maintains modality purity within individual training batches by enforcing strict image-video alternation. To accommodate native-resolution video processing with dynamic lengths, we implement adaptive frame sampling: full temporal retention when frame count $F < F_{max}$, and uniform subsampling to $F_{max}$ frames when exceeded. With the predefined token constraints $(L_{min}, L_{max})$ and the calculated frame length $F$, all frames are subsequently resized within these computed bounds while preserving original aspect ratios.

## 2.3 UniViTAR as a Vision Encoder for MLLMs.

In this section, we introduce a simple strategy for constructing an effective native resolution MLLM based on the UniViTAR series. The common and industry-validated Vision-Language Models (VLMs) paradigm typically combines pretrained visual backbones with large language models, followed by multimodal training on a rich mixture of vision language tasks. To ensure fair comparison and minimize bias, we adhere to this established configuration. Specifically, we employ UniViTAR as the vision encoder and employ Qwen2.5-1.5B [5] as the large language model. Following established practices [17], we implement a three-layer *MLP* with pre-normalization and a $2\times$ pixel-unshuffle operation [18] along the width dimension as the vision-language adapter to bridge the visual and linguistic modalities. For native-resolution modeling, we identify two primary challenges. On one hand, due to the varying lengths of input samples, the boundary between vision and language tokens is not fixed. To enhance "modality isolation", we introduce specialized prompts, known as *Boundary Markers*, such as *<image_start>* and *<image_end>*, at the beginning and end of the vision token sequence. On the other hand, 2D-to-1D flattening of vision tokens may compromise the information of the height-width ratio. To mitigate this, we incorporate *Line Anchors*, such as <line-*idx*>, into the vision tokens, where *idx* denotes the corresponding vertical positions in the original patchified image, thereby potentially strengthening positional awareness in compressed tokens. For a vision token sequence of length $hw$, the original arrangement $x^{1,1}, ..., x^{1,w}, ..., x^{2,w}, ..., x^{h,w}$ is transformed as:

$$<image\_start>, x^{1,1}, \ldots, x^{1,w}, <line-1>, x^{2,1}, \ldots, x^{h,w}, <line-h>, <image\_end> \quad (2)$$

Notably, these added markers are string-based identifiers rather than special tokens of the tokenizer. To systematically evaluate multimodal comprehension capabilities, we adopt a dual-stage training paradigm motivated by established methodologies in vision-language alignment like [19, 20].

# 3 Experiments

## 3.1 Training Recipe

***Data Details.*** We collect public accessible image-text pairs and build our Merged-1B dataset, which is composed of DataComp-1B [21], COYO [22], LAION-2B [23], LAION-400M [24], DFN-2B [22], CC12M [25] and CC3M [26]. Moreover, to further enhance the video feature extraction capabilities of UniViTAR, we meticulously constructed a dataset Merged-65M of roughly 65 million samples by randomly selecting video clips from three public accessible video datasets, *i.e.*, Panda-70M [27], WebVid-10M [28], and InternVid-10M-FLT [29]. We refer to the combined image and video data mentioned above as Merged-1.1B. The detailed data composition is summarized in the Appendix.

***Hyperparameter Details.*** The detailed hyperparameter configurations for each training stage are presented in the Appendix. As tabulated, we utilize a progressive reduction of the peak learning rate in correlation with increasing visual backbone scale to ensure optimal training stability. Notably, the learning rate of text branch in Stage 2 remains consistently one-tenth of the visual component throughout this phase. To enhance training efficiency, we integrated the DeepSpeed library [30] by employing ZeRO optimizer sharding [31], gradient checkpointing [32], and flash attention [33].

## 3.2 Results on Zero-shot Image Classification & Retrieval

***Evaluation Setup.*** Our evaluation protocol encompasses both zero-shot classification and cross-modal retrieval tasks. For zero-shot classification, we conduct evaluation on ImageNet [34] and its established variants [35, 36, 37, 38, 39]. Each class is represented by multiple text prompts curated from [11, 40]. The *Top-1* accuracy is utilized to evaluate the model performance. For cross-modal retrieval assessment, we adopt the benchmark protocols defined in [41], evaluating on Flickr [42] and MS-COCO [43] using their official partitions. The retrieval paradigm involves bidirectional image-text matching, namely image-to-text retrieval and text-to-image retrieval tasks.

***Results Comparison and Analysis.*** Table 3 demonstrates the exceptional performance of our model at comparable parameter scales. As the model size increases from 0.3B to 1.4B, the average zero-shot classification accuracy across six benchmarks exhibits a progressive improvement trend, rising from 80.5% to 81.9% and further to 83.4%. Notably, all models of varying scales employ identical training samples and strategies, with this performance enhancement attributed to *parameter scaling* effects—a finding consistent with established scaling laws in transformer-related research. As detailed in the table, our UniViTAR-1B shows superior performance despite utilizing a smaller training corpus, outperforming its counterparts with more parameters, such as InternViT-6B [12] and EVA-8B [44]. We posit that this advantage stems from two key factors: optimized model atchitecture and training strategy, and preservation of native input resolution, which generates higher-quality visual tokens.

## 3.3 Results on Zero-shot Video Classification & Retrieval

***Evaluation Setup.*** We evaluate the zero-shot video classification performance on three popular benchmarks as K-400 [50], UCF-101 [51] and HMDB51 [52], using the class names as text prompts. Also, we evaluate the zero-shot video-text retrieval performance on ActivityNet [53], MSR-VTT [54] and MSVD [55]. Following [15, 16], for each video in the 1K version of the test split, we sample one sentence from every set of 20 sentences for MSR-VTT. Following [56], we concatenate the multiple descriptions to form a paragraph and perform a paragraph-to-video retrieval on ActivityNet. All videos are sampled with a dynamic frame rate, with each frame dynamically resized to maintain the original aspect ratio while ensuring the total token within the range of 576 to 16,384.

***Results Comparison and Analysis.*** Table 4 shows the performance of our UniViTAR series models on video benchmarks across comparable parameter scales. As the model size scales from 0.3B to 1B, UniViTAR exhibits consistent performance gains on video benchmarks, with average zero-shot classification metrics improving from 68.0 to 69.0. When compared to models trained on image-caption data under similar parameter scales, UniViTAR achieves notable improvements. These advancements can be attributed to two key design choices: (1) preserving the aspect ratio of each frame to retain the original semantic information of visual content, and (2) employing dynamic video frame sampling to effectively capture detailed temporal information. However, when compared to the models trained exclusively on video-caption data, UniViTAR still has room for improvement compared to some of the latest models [57, 58, 16], as shown in the Table 4 with gray color.

Table 3: **Evaluation of zero-shot performance on various image benchmarks**. The symbol ⌀ indicates that the image-caption data used by the corresponding method is not publicly available.

| Method | Data Source | Res. | Overall | ImageNet Variants | | | | | | Overall | Flickr | | COCO | |
| --- | --- | --- | --- | --- | --- | --- | --- | --- | --- | --- | --- | --- | --- | --- |
| | | | | IN-1K | IN-A | IN-R | IN-V2 | IN-S | O-Net | | T→I | I→T | T→I | I→T |
| CLIP-L [11] | WIT400M ⌀ | 224 | 72.1 | 75.5 | 70.8 | 87.8 | 69.8 | 59.6 | 68.9 | 60.8 | 65.0 | 85.2 | 36.5 | 56.3 |
| OpenCLIP-L [45] | DataComp1B | 224 | 75.7 | 79.2 | 69.6 | 90.8 | 72.1 | 68.0 | 74.3 | 67.9 | 73.4 | 89.0 | 45.7 | 63.3 |
| MetaCLIP-L [46] | CC-2.5B ⌀ | 224 | 76.6 | 79.2 | 72.3 | 92.1 | 72.6 | 69.0 | 74.6 | 69.5 | 76.4 | 90.1 | 47.1 | 64.4 |
| DFN-L [47] | DFN5B ⌀ | 224 | 77.1 | **82.2** | 67.5 | 91.8 | 75.7 | 70.4 | 74.8 | 69.8 | 75.5 | 89.6 | 48.6 | 65.6 |
| EVA02-L [44] | Merged-2B | 336 | 77.5 | 79.8 | 76.2 | 92.7 | 73.0 | 68.1 | 74.9 | 69.9 | 78.0 | 89.6 | 47.9 | 64.2 |
| CLIPAv2-L [48] | DataComp1B | 336 | 78.1 | 80.3 | 77.7 | 93.3 | 73.5 | 70.9 | 73.1 | 69.5 | 74.6 | 90.4 | 47.2 | 65.6 |
| SigLIP-L [6] | WebLI10B-En ⌀ | 384 | 79.4 | 82.1 | 76.6 | **95.1** | 75.9 | 73.6 | 72.8 | 75.2 | 81.4 | 93.7 | 53.9 | **71.9** |
| UniViTAR-0.3B | Merged-1B | Native | **80.6** | 81.5 | **84.1** | 93.9 | 75.1 | 69.7 | **79.1** | **76.3** | **84.0** | **95.1** | **54.7** | 71.2 |
| OpenCLIP-H [45] | LAION2B-en | 224 | 72.3 | 78.0 | 59.4 | 89.3 | 70.9 | 66.6 | 69.4 | 68.7 | 75.5 | 89.5 | 46.5 | 63.4 |
| MetaCLIP-H [46] | CC-2.5B ⌀ | 224 | 78.4 | 80.5 | 75.3 | 93.4 | 74.2 | 70.5 | 76.4 | 71.3 | 78.3 | 91.8 | 48.8 | 66.2 |
| CLIPAv2-H [48] | DataComp1B | 336 | 80.8 | 81.8 | 82.7 | 94.4 | 75.6 | 72.8 | 77.4 | 70.8 | 76.3 | 90.3 | 49.2 | 67.2 |
| DFN-H [47] | DFN5B ⌀ | 378 | 80.5 | **84.4** | 79.6 | 93.8 | **78.3** | 73.2 | 73.4 | 75.9 | 82.0 | 94.0 | **55.6** | 71.9 |
| SigLIP-SO [6] | WebLI10B-En ⌀ | 384 | 81.7 | 83.1 | 82.5 | **95.8** | 77.2 | **74.5** | 77.0 | 76.0 | 83.0 | 94.3 | 54.2 | **72.4** |
| UniViTAR-0.6B | Merged-1B | Native | **82.1** | 82.3 | **86.8** | 94.9 | 76.1 | 71.6 | **81.1** | **76.6** | **84.1** | **95.5** | 55.4 | 71.7 |
| OpenCLIP-g [45] | LAION2B-en | 224 | 73.0 | 78.5 | 60.9 | 90.2 | 71.6 | 67.5 | 69.1 | 71.1 | 77.7 | 91.4 | 48.8 | 66.4 |
| OpenCLIP-G [45] | LAION2B-en | 224 | 76.2 | 80.1 | 69.3 | 92.1 | 73.6 | 68.9 | 72.8 | 72.8 | 79.6 | 92.9 | 51.4 | 67.4 |
| EVA01-g [49] | Merged-2B | 224 | 76.9 | 79.4 | 74.2 | 92.5 | 72.1 | 68.1 | 74.9 | 72.3 | 79.0 | 91.7 | 50.3 | 68.2 |
| EVA02-E [44] | Merged-2B | 336 | 80.9 | 82.0 | 82.2 | 94.6 | 75.6 | 71.6 | 79.4 | 73.2 | 78.9 | 94.1 | 51.1 | 68.7 |
| CLIPAv2-G [48] | DataComp1B | 336 | 82.7 | 83.1 | 86.0 | 95.4 | 77.3 | **74.5** | 79.7 | 72.2 | 78.3 | 92.2 | 50.4 | 67.8 |
| InternViT-6B [12] | InternVL-5B | 224 | 82.5 | 83.2 | 83.8 | **95.7** | 77.3 | 74.3 | 80.6 | 75.3 | 81.7 | 94.7 | 54.1 | 70.6 |
| EVA-8B [49] | Merged-2B | 224 | 82.9 | **83.5** | 85.2 | 95.3 | **77.7** | 74.3 | 81.2 | 74.9 | 80.8 | **95.6** | 53.0 | 70.3 |
| UniViTAR-1B | Merged-1B | Native | **83.5** | 82.9 | **89.1** | **95.7** | 77.3 | 73.4 | **82.8** | **76.3** | 83.5 | 95.1 | 55.3 | 71.3 |

Table 4: **Evaluation of zero-shot performance on various video benchmarks**. The symbol † signifies that the reported metrics are based on our own evaluations.

| Method | Type | Res. | Frames | Overall | Classification | | | Overall | ANet | | MSR-VTT | | MSVD | |
| --- | --- | --- | --- | --- | --- | --- | --- | --- | --- | --- | --- | --- | --- | --- |
| | | | | | K400 | UCF | HMDB | | V→T | T→V | V→T | T→V | V→T | T→V |
| †OpenCLIP-L [45] | Image | 224 | 16 | 58.4 | 61.5 | 69.2 | 44.5 | 41.0 | 32.0 | 34.2 | 30.1 | 37.5 | 63.7 | 48.5 |
| †DFN-L [47] | Image | 224 | 16 | 56.4 | 56.8 | 67.7 | 44.8 | 40.4 | 31.6 | 34.1 | 32.1 | 35.2 | 61.9 | 47.7 |
| †EVA02-L [44] | Image | 336 | 16 | 64.4 | 64.4 | 76.0 | 52.8 | 44.7 | 35.8 | 37.2 | 35.4 | 39.7 | 69.1 | 51.0 |
| †SigLIP-L [6] | Image | 384 | 16 | 64.8 | 64.2 | 79.2 | 50.9 | 45.3 | 34.3 | 35.8 | 35.7 | 40.0 | 73.0 | **53.0** |
| ViCLIP-L [29] | Video | 224 | 8 | - | 64.8 | - | - | 41.2 | 24.0 | 15.1 | 41.3 | 42.4 | 75.1 | 49.1 |
| InterVideo-L [15] | Video | 224 | 16 | - | 64.3 | 80.5 | - | 42.2 | 31.4 | 30.7 | 39.6 | 40.7 | 67.5 | 43.4 |
| UMT-L [59] | Video | 224 | 16 | - | - | - | - | 47.7 | 39.4 | 41.9 | 38.6 | 42.6 | 74.5 | 49.0 |
| UniViTAR-0.3B | Image&Video | Native | 2∼32 | **68.0** | **66.0** | **82.6** | **55.4** | **53.9** | **47.9** | **49.9** | **48.0** | **48.8** | **77.8** | 50.7 |
| †OpenCLIP-H [45] | Image | 224 | 16 | 62.0 | 61.7 | 72.5 | 51.6 | 43.5 | 36.1 | 38.9 | 34.5 | 38.9 | 63.3 | 49.4 |
| †DFN-H [47] | Image | 378 | 16 | 62.9 | 63.8 | 76.7 | 48.2 | 46.2 | 39.7 | 42.9 | 36.1 | 39.6 | 66.6 | 52.4 |
| †SigLIP-SO [6] | Image | 384 | 16 | 67.3 | 66.8 | **83.0** | 52.1 | 47.5 | 36.6 | 39.3 | 37.5 | 41.1 | 75.5 | **54.7** |
| TVTSV2-H [60] | Video | 224 | 12 | 63.2 | 59.6 | 78.0 | 52.1 | - | - | - | - | 41.3 | - | - |
| UniViTAR-0.6B | Image&Video | Native | 2∼32 | **68.6** | **67.6** | 82.9 | **55.2** | **54.9** | **48.7** | **51.5** | **48.6** | **50.2** | 75.8 | 54.3 |
| †OpenCLIP-g [45] | Image | 224 | 16 | 63.1 | 61.5 | 76.6 | 51.1 | 44.4 | 36.8 | 39.8 | 36.4 | 39.2 | 64.3 | 50.1 |
| †OpenCLIP-G [45] | Image | 224 | 16 | 64.2 | 63.2 | 76.2 | 53.4 | 46.0 | 36.7 | 41.4 | 36.9 | 41.8 | 67.5 | 51.5 |
| †EVA01-g [49] | Image | 224 | 16 | 62.8 | 63.4 | 72.1 | 52.9 | 45.5 | 37.0 | 40.1 | 37.2 | 40.1 | 67.6 | 50.8 |
| InternViT-6B [12] | Image | 224 | 8 | - | 69.1 | - | - | - | - | - | 42.4 | 46.3 | - | - |
| UniViTAR-1B | Image&Video | Native | 2∼32 | **69.0** | 68.6 | 81.0 | **57.3** | 54.0 | 47.8 | 49.6 | 48.3 | 47.6 | 75.5 | 55.2 |
| VideoCoCa-g [57] | Video | 224 | 8 | 72.4 | 72.0 | 86.6 | 58.7 | 39.0 | 33.0 | 34.5 | 64.7 | 34.4 | 33.0 | 34.5 |
| VideoPrism-g [58] | Video | 288 | 16 | - | 76.4 | - | - | - | 50.3 | 52.7 | 51.7 | 52.7 | - | - |
| InternVideo2-6B [16] | Video | 224 | 8 | - | - | - | - | 62.0 | 56.5 | 63.2 | 53.7 | 55.9 | 83.1 | 59.3 |

## 3.4 Results on Image Classification by Linear Probing

Following common prectices [12, 61], we assess the performance of UniViTAR family as off-the-shelf backbones on image classifications. Specifically, we train a linear classifier on the last feature layer with a frozen backbone on ImageNet-1K [34] and evaluate the performance on the validation set and other ImageNet variants [62, 35, 36, 37, 38]. In addition, we also report the classification performance with attentive probing setting as used in [61], which adopts a cross-attention layer with random initialized queries. Table 5 represents the downstream classification performance of our models. First, as the model size increases, the average performance across six benchmarks demonstrates consistent improvement. Second, we observe that the attentive probing performance shows stable improvements over linear probing. Furthermore, compared to public methods, our UniViTAR family shows superior performance across various parameter scales.

Table 5: **Evaluation of classification performance on various image benchmarks**. The † signifies that the reported metrics are based on our own evaluations.

| Method | Classifier | Res. | Overall | ImageNet Variants | | | | | |
|---|---|---|---|---|---|---|---|---|---|
| | | | | IN-1K | IN-Real | IN-V2 | IN-A | IN-R | IN-S |
| CLIP-L [11] | Linear | 336 | - | 85.3 | 88.8 | 75.8 | - | - | - |
| SigLIP-L [6] | Attentive | 224 | - | 86.5 | - | - | - | - | - |
| AIMv2-L [61] | Attentive | 224 | - | 86.6 | - | - | - | - | - |
| UniViTAR-0.3B | Linear | Native | 83.0 | 87.6 | 90.3 | 79.5 | **84.1** | 90.6 | 66.0 |
| UniViTAR-0.3B | Attentive | Native | **83.3** | **87.7** | **90.5** | **79.8** | 83.8 | **91.1** | **66.8** |
| CLIP-H [11] | Linear | 224 | - | 84.4 | 88.4 | 75.5 | - | - | - |
| †DFN-H [47] | Linear | 378 | 81.6 | 87.3 | 90.4 | 78.8 | 74.8 | 90.3 | 68.3 |
| SigLIP-SO [6] | Attentive | 384 | - | 87.3 | - | - | - | - | - |
| AIMv2-H [61] | Attentive | 224 | - | 87.5 | - | - | - | - | - |
| UniViTAR-0.6B | Linear | Native | 84.4 | 88.2 | 90.6 | 80.6 | 87.1 | 92.0 | 68.0 |
| UniViTAR-0.6B | Attentive | Native | **84.8** | **88.3** | **90.7** | **81.0** | **87.3** | **92.5** | **68.8** |
| OpenCLIP-G [45] | Linear | 224 | 78.5 | 86.2 | 89.4 | 77.2 | 63.8 | 87.8 | 66.4 |
| DINOv2-g [13] | Linear | 224 | 78.6 | 86.5 | 89.6 | 78.4 | 75.9 | 78.8 | 62.5 |
| EVA01-g [49] | Linear | 224 | 79.1 | 86.5 | 89.3 | 77.4 | 70.5 | 87.7 | 63.1 |
| AIMv2-1B [61] | Attentive | 224 | - | 88.1 | - | - | - | - | - |
| InternViT-6B [12] | Linear | 224 | 82.5 | 88.2 | 90.4 | 79.9 | 77.5 | 89.8 | 69.1 |
| EVA-8B [49] | Linear | 224 | - | 88.5 | - | - | - | - | - |
| UniViTAR-1B | Linear | Native | **86.0** | 88.9 | 90.8 | 81.5 | 90.1 | **94.0** | **70.7** |
| UniViTAR-1B | Attentive | Native | **86.0** | **89.2** | **91.0** | **81.7** | 90.1 | 93.6 | 70.6 |

## 3.5 Results on Dense prediction.

In this section, we evaluate the dense prediction performance of our UniViTAR family by transferring to semantic segmentation. Following [12, 63], we fine-tune a decoder with freezing backbones under two different structures, *i.e.*, Linear and UperNet. Linear decoder transforms the dimension of one single layer visual feature to number of semantic classes, while the UperNet decoder employs PPM and FPN to integrates multi-scale features. Experiments are conducted on the ADE20K [64] dataset. In terms of data preprocessing, we employed the same fixed-resolution input and data augmentation strategies as those used in InternViT [12]. Corresponding results are shown in Table 6. We can observe a performance gap between these two types of decoder, this can be understand that UperNet has significantly more trainable parameters than Linear decoder. Taking UniViTAR-0.6B as an example, Linear decoder has a parameter count of 0.2M, whereas UperNet contains approximately 200M parameters. Notably, our UniViTAR Family demonstrates an obvious performance advantage compared with existing state-of-the-art vision encoders. Under the setting of Linear decoder, our UniViTAR-1B achieves a performance of 45.4 mIoU, which is $+6.1$ points over OpenCLIP-G [45] and $+10.8$ points over ViT-22B [10]. In the case of UperNet decoder, our UniViTAR-1B reaches 56.2 mIoU, also surpassing larger parameter-scale model like InternViT-6B [12].

Table 6: **Evaluation of semantic segmentation on ADE20k dataset with frozen backbones.**

| Method | CropSize | mIoU$^{Linear}$ | mIoU$^{UperNet}$ |
|---|---|---|---|
| CLIP-L [11] | - | 39.0 | - |
| SigLIP-SO [6] | - | 40.8 | - |
| †DFN-H [47] | - | 41.3 | - |
| OpenCLIP-G [45] | $512^2$ | 39.3 | |
| InternViT-6B [12] | $504^2$ | **47.2** | 54.9 |
| ViT-22B [10] | $504^2$ | 34.6 | 52.7 |
| UniViTAR-0.3B | $504^2$ | 40.7 | 54.6 |
| UniViTAR-0.6B | $504^2$ | 42.9 | 55.1 |
| UniViTAR-1B | $504^2$ | 45.4 | **56.2** |

## 3.6 Results on Multimodal Understarding

***Evaluation Setup.*** To assess the potential of multimodal understanding, we employ a dual-stage training paradigm, similar to common practices [19, 20]. In the pretraining stage, we train the projector with a learning rate of $1e^{-3}$ using a merged 2.5M dataset comprised of LLaVA-CC3M-Pretrain [17], ALLaVA-Caption [19], ShareGPT4V-PT [20]. In the fine-tuning stage, we unfreeze the whole model, and train it with a learning rate of $1e^{-5}$, using the high-quality instruction-tuning

dataset LLaVA1.5-Finetune [65]. Note that the native-resolution strategy of *Boundary Markers* and *Line Anchors* are only applied in the fine-tuning stage. All evaluations are conducted using VLMEvalKit [66], assessing performance across 16 popular benchmarks, including GQA [67], DocVQA [68], InfoVQA [69], ScienceQA [70], TextVQA [71], VizWiz [72], OCRVQA [73], OCR-Bench [74], MME [75], MMMU [76], SEEDBench_IMG [77], MathVista_MINI [78], AI2D [79], HallusionBench [80], POPE [81], HRBench4K [82].

***Results Comparison and Analysis.*** As illustrated in Table 7, under exactly the same training data and training strategy, the proposed UniViTAR surpasses various state-of-the-art vision encoders [47, 61, 6] on numerous multimodal understanding benchmarks. Notably, UniViTAR demonstrates exceptional capabilities in scenarios involving dense information, such as document parsing [68], graphic parsing [69], and high-resolution tasks [82]. We argue that the native resolution plays a crucial role in achieving outstanding performance in these areas, ensuring minimal loss of image information. We also assess the effectiveness of the proposed strategy of *Boundary Markers* and *Line Anchors* on 0.6B model size, as demonstrated in Table 7, which highlights their impact.

Table 7: **Evaluation of multimodal understanding on various vision-language benchmarks**. Note the superscript △ represents model with *Boundary Markers* and *Line Anchors*.

| Benchmarks | SigLIP-L [6] | DFN-H [47] | AIMv2-H [61] | SigLIP-SO [6] | UniViTAR | | | |
| | | | | | 0.3B△ | 0.6B | 0.6B△ | 1B△ |
| Resolution | 378 | 378 | 448 | 384 | Native | | | |
| GQA$_{TestDev}$ | 61.5 | 60.6 | 61.5 | 61.0 | 60.8 | 58.2 | 60.3 | 61.2 |
| DocVQA$_{VAL}$ | 30.8 | 25.9 | 36.2 | 32.0 | 47.7 | 46.3 | 48.2 | 47.0 |
| InfoVQA$_{VAL}$ | 22.7 | 22.1 | 25.8 | 23.2 | 27.8 | 28.0 | 28.5 | 27.5 |
| ScienceQA$_{VAL}$ | 63.6 | 62.7 | 64.8 | 66.4 | 64.5 | 65.0 | 63.6 | 65.3 |
| TextVQA$_{VAL}$ | 48.0 | 41.7 | 53.2 | 50.9 | 50.8 | 52.0 | 50.7 | 52.0 |
| VizWiz | 30.5 | 28.5 | 30.3 | 30.8 | 29.1 | 29.8 | 29.4 | 29.3 |
| OCRVQA | 31.2 | 32.0 | 31.0 | 30.9 | 32.2 | 31.6 | 32.2 | 32.1 |
| OCRBench | 35.2 | 30.6 | 22.4 | 36.0 | 33.6 | 37.0 | 36.9 | 36.4 |
| MME | 59.3 | 62.6 | 59.8 | 60.0 | 57.9 | 58.6 | 59.0 | 60.7 |
| MMMU$_{VAL}$ | 35.9 | 34.2 | 37.1 | 35.4 | 36.1 | 38.7 | 36.6 | 37.0 |
| SEEDBench_IMG | 70.0 | 70.3 | 70.9 | 71.2 | 68.2 | 67.8 | 68.5 | 69.3 |
| MathVista_MINI | 28.6 | 29.9 | 30.1 | 29.6 | 27.5 | 27.9 | 28.5 | 28.7 |
| AI2D$_{Test}$ | 60.5 | 58.3 | 60.4 | 60.6 | 58.0 | 57.7 | 58.7 | 59.3 |
| HallusionBench | 56.8 | 56.6 | 53.9 | 54.8 | 57.0 | 55.2 | 57.8 | 54.1 |
| POPE | 87.2 | 88.0 | 85.4 | 87.7 | 87.1 | 88.1 | 87.9 | 88.1 |
| HRBench4K | 39.9 | 39.5 | 44.5 | 45.0 | 44.1 | 43.6 | 46.1 | 46.4 |
| Average | 47.6 | 46.5 | 48.0 | 48.5 | 48.9 | 49.1 | 49.5 | 49.6 |

## 3.7 Ablation Study

### 3.7.1 Robustness verification of resolution mode.

This section analyzes the influence of three resolution modes: fixed resolution, native aspect ratio, and native resolution in Figure 3. For fixed resolution mode, we resize the shorter edge of each image to a predefined size $S$ and apply *CenterCrop* to ensure the sequence length strictly equals $(S/14)^2$, where 14 represents the patch size. Increasing $S$ proportionally extends the sequence length. In native aspect ratio mode, we scale images while preserving their original aspect-ratio, ensuring that $wh/14^2$ approximates the target sequence length. We evaluate 12 sequence lengths ranging from 256 to 16K tokens with our 0.6B model. The experimental results reveal three key findings: *1)* performance initially improves then declines with increasing sequence length under both fixed and native aspect ratio modes, peaking at 1024∼4096 tokens. *2)* native aspect ratio consistently outperforms fixed resolution, indicating that preserving original aspect ratios retains better image information. *3)* native aspect ratio occasionally surpasses native resolution performance at certain lengths.

### 3.7.2 Verification of the effectiveness of training strategies.

As introduced in the method, we categorize the training strategy into four distinct stages, based on resolution modes (fixed or native), visual data modalities (image or video), and model training parameters (trainable or freeze). As shown in Figure 4, for *zero-shot image classification* (left), we show that S1, S2, and S3 exhibit progressive performance improvements, while S4 maintains comparable accuracy despite incorporating alternating training. In contrast, for *zero-shot video classification* (right), S1 and S2 show minimal performance variation, with dynamic-resolution

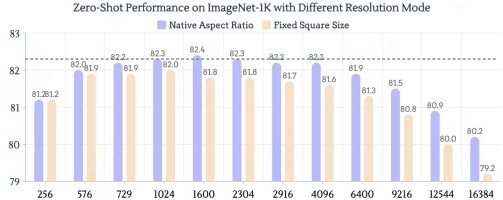
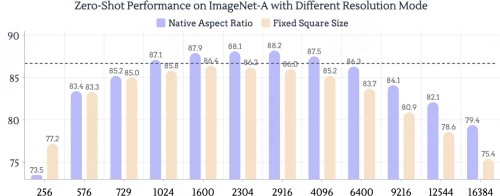

Figure 3: **Performance comparison of different resolution modes as the length of the vision sequence increases**. The black dashed line shows the performance when using native resolution.

training in S3 significantly boosting video capabilities, followed by further enhancements in S4 through image-video training. This demonstrates that dynamic-resolution training enables model to process more native visual sequences, while the final unified training stage equips the model with generalized capabilities for handling diverse visual modalities. Notably, these findings remain highly consistent across the UniViTAR-0.3B/0.6B/1B model family.

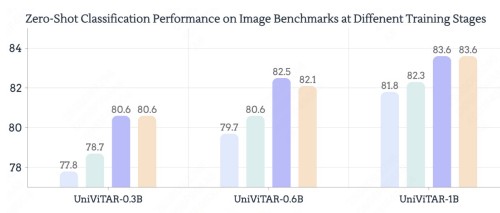
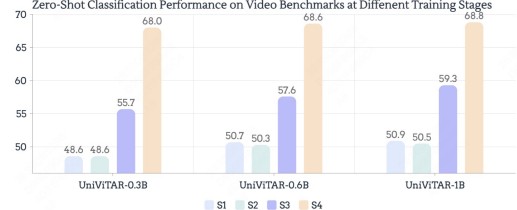

Figure 4: **Average performance improvement illustration across different training stages**.

### 3.8 Verification of the effectiveness of hybrid training with DINOv2.

To investigate the performance benefits of incorporating DINOv2 as a distillation branch in Stage-1 training, we performed a comprehensive empirical study using the UniViTAR-0.3B model trained from scratch on 3B image-text pairs. Checkpoints were evaluated at regular intervals throughout training in Table 8. Our experiments revealed that DINOv2 distillation significantly accelerates early-stage convergence, after only 0.1B samples, distillation improved zero-shot ImageNet-1K performance by 17.3 points. Although this gain gradually diminishes as training progresses, it remains observable at later stages. More importantly, the final model trained with DINOv2 distillation achieves an average improvement of 2.1 points across six zero-shot classification benchmarks compared to the baseline without distillation (Table 9). These results demonstrate that DINOv2 distillation not only speeds up early convergence but also enhances the final model performance.

Table 8: **The performance gains of hybrid training on ImageNet-1K as data increases**.

| Model | DINOv2 | 0.1B | 0.5B | 1.0B | 1.5B | 2.0B | 2.5B | 3.0B |
|---|---|---|---|---|---|---|---|---|
| UniViTAR-0.3B | No | 26.88 | 63.71 | 67.80 | 69.75 | 72.32 | 74.88 | 75.72 |
| UniViTAR-0.3B | Yes | 44.18 | 68.15 | 71.24 | 73.17 | 74.71 | 76.40 | 77.33 |
| Δ | - | 17.30 | 4.44 | 3.44 | 3.42 | 2.39 | 1.52 | 1.61 |

Table 9: **Zero-shot classification performance of hybrid training with DINOv2**.

| Model | DINOv2 | Avg. | IN-1K | IN-A | IN-R | IN-V2 | IN-S | O-Net |
|---|---|---|---|---|---|---|---|---|
| UniViTAR-0.3B | No | 70.73 | 75.72 | 58.76 | 87.98 | 68.22 | 62.95 | 70.78 |
| UniViTAR-0.3B | Yes | 72.84 | 77.33 | 63.55 | 89.81 | 70.40 | 65.65 | 70.31 |
| Δ | - | 2.11 | 1.61 | 4.79 | 1.83 | 2.18 | 2.70 | -0.47 |

## 4 Conclusion

In this work, we introduce UniViTAR, a family of homogeneous vision foundation models tailored for unified visual modality and native-resolution scenarios in the era of multimodal. By integrating advanced architectural upgrades, resolution curriculum learning, visual feature distillation, and inter-batch modality adaptation, UniViTAR achieves significant improvements across diverse tasks, spanning image/video zero-shot classification/retrieval, dense prediction accuracy, and vision-language model transfer performance. Notably, all models are trained exclusively on public-accessible datasets, where we observe consistent performance gains with parameter scaling from 0.3B to 1.4B. We hope that our UniViTAR offers the community a versatile framework for advancing multimodal research.

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

## A The Overview of UniViTAR Pipeline

The proposed UniViTAR processes visual input at its native resolution, and also supports scaling the resolution down or up while maintaining the aspect ratio or resizing to certain square size to accommodate different application scenarios, such as higher computational efficiency or finer-grained visual details. By treating video inputs as temporally extended image sequences, the framework uniformly produces longer variable-length visual token sequences.

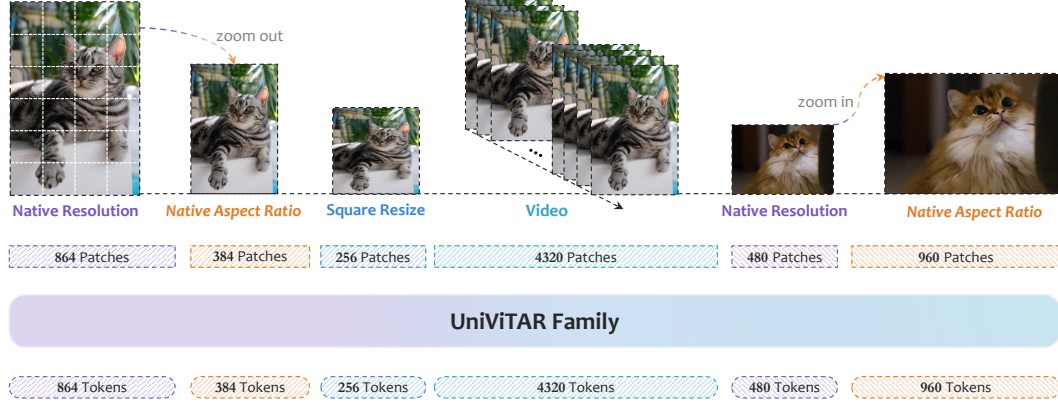

Figure 5: **The brief illustration of UniViTAR family pipeline**.

## B More Details of Training Recipe

***Data Details.*** We collect public accessible image-text pairs and build our Merged-1B dataset, which is composed of DataComp-1B [21], COYO [22], LAION-2B [23], LAION-400M [24], DFN-2B [22], CC12M [25] and CC3M [26]. Moreover, to further enhance the video feature extraction capabilities of UniViTAR, we meticulously constructed a dataset Merged-65M of roughly 65 million samples by randomly selecting video clips from three public accessible video datasets, *i.e.*, Panda-70M [27], WebVid-10M [28], and InternVid-10M-FLT [29]. We refer to the combined image and video data mentioned above as Merged-1.1B. The detailed data composition is summarized in the Table 10.

Table 10: **Details of the training data for UniViTAR**. Note that Merged-1B and Merged-65M correspond to image and video modality respectively.

| Dataset | Source | Language | Samples | Total | Percentage | Used by |
|---|---|---|---|---|---|---|
| Merged-1B | DataComp-1B [21] | En | 408M | | 37.7% | |
| | COYO [22] | En | 248M | | 22.9% | |
| | LAION-2B [23] | En | 213M | | 19.7% | |
| | DFN-2B [47] | En | 154M | 1.08B | 14.3% | Stage 1~4 |
| | LAION-400M [24] | En | 52.7M | | 4.9% | |
| | CC12M [25] | En | 2.94M | | 0.3% | |
| | CC3M [26] | En | 2.32M | | 0.2% | |
| Merged-65M | Panda-70M [27] | En | 52.1M | | 80.2% | |
| | WebVid-10M [28] | En | 6.53M | 65M | 10.0% | Stage 4 |
| | InternVid-10M-FLT [29] | En | 6.31M | | 9.8% | |

***Hyperparameter Details.*** The detailed hyperparameter configurations for each training stage are presented in the Table 11. As tabulated, we utilize a progressive reduction of the peak learning rate in correlation with increasing visual backbone scale to ensure optimal training stability. Notably, the learning rate of text branch in Stage 2 remains consistently one-tenth of the visual component throughout this phase. To enhance training efficiency, we integrated the DeepSpeed library [30] by employing ZeRO optimizer sharding [31], gradient checkpointing [32], and flash attention [33]. Note all experiments are conducted on H800 GPUs.

Table 11: **Detailed training hyperparameter of UniViTAR family**. Note that the symbol of $\rightarrow$ represents the peak learning rate and the minimum learning rate in the LR schedule.

| | Stage 1 | Stage 2 | Stage 3 | Stage 4 |
|---|---|---|---|---|
| Vision Encoder *Init.* | Xavier *init.* [83] | from Stage-1 | from Stage-2 | from Stage-3 |
| Text Encoder *Init.* | LLama [12] | LLama [12] | from Stage-2 | from Stage-3 |
| Input Resolution | $224 \times 224$ | $224 \times 224$ | **Native** | **Native** |
| Token Range | 256 | 256 | $64 \sim 16K$ | $64 \sim 16K$ |
| Global Batch Size | 32768 | 32768 | 32768 | $\sim$26K(Image), $\sim$4K(Video) |
| Patch Dropout | 0.5 | 0.0 | 0.5 | 0.5 |
| Warmup Steps | 2000 | 2000 | 2000 | 1000 |
| Optimizer | AdamW | AdamW | AdamW | AdamW |
| LR Schedule | Cosine Decay | Cosine Decay | Cosine Decay | Cosine Decay |
| *0.3B* | $1e^{-3} \rightarrow 1e^{-6}$ | $1e^{-5} \rightarrow 0$ | $1e^{-5} \rightarrow 0$ | $4e^{-6} \rightarrow 0$ |
| *0.6B* | $1e^{-3} \rightarrow 1e^{-6}$ | $1e^{-5} \rightarrow 0$ | $1e^{-5} \rightarrow 0$ | $4e^{-6} \rightarrow 0$ |
| *1B* | $8e^{-4} \rightarrow 1e^{-7}$ | $6e^{-6} \rightarrow 0$ | $6e^{-6} \rightarrow 0$ | $2e^{-6} \rightarrow 0$ |
| Train Dataset | Merged-1B | Merged-1B | Merged-1B | Merged-1B, Merged-65M |
| Seen Samples | 12B | 1B | 1B | 0.6B |

# C More Experimental Results & Ablation Study

## C.1 Verification of the effectiveness of image-video alternative strategy.

To validate the efficacy of the alternating image-video training strategy, we conducted initial experiments with 100M image-text pairs and 10M video-text pairs. Note that the image-to-video data ratio is approximately 10:1, consistent with the ratio used in stage 4 of the UniViTAR series. We trained a UniViTAR-0.3B model for 3 epochs, comparing mixed training and alternating training strategies. As shown in Table 12, the alternating training strategy outperforms the mixed strategy across key image and video benchmark metrics, demonstrating its effectiveness in enhancing visual representation learning. This performance gain can be attributed to the increased training difficulty arising from the unification of data modalities within each batch.

Table 12: **Zero-shot classification performance of image-video training strategy**.

| Strategy | ImageNet-1K | ImageNet-A | K400 | UCF101 |
|---|---|---|---|---|
| Batch-Mixed | 70.46 | 45.89 | 58.82 | 75.15 |
| Batch-Alternative | 71.25 | 48.60 | 61.01 | 77.66 |

## C.2 Verification of the effectiveness of native resolution for video.

In this section, we conduct an ablation study to explore the role of native resolution in video data processing. We dynamically sample a maximum of 32 frames (denoted as $F$) for each video clip. For frames exceeding the sequence length limit, we resize them while preserving their native aspect ratio to a smaller resolution. We evaluate 15 maximum video sequence length, ranging from 1024 to 65,536, and test the zero-shot classification performance of UniViTAR-0.6B on the K400 dataset. Note that the minimum video sequence length is fixed to 576. As shown in Figure 6, the performance initially improves and then stabilizes as the sequence length limit increases, reaching a plateau at length 10,240. We attribute this to the fact that, with 32 sampled frames, a sequence length of 10,240 corresponds to a resolution of $490 \times 256$, enabling most frames in K400 to retain their native resolution during data processing. This finding underscores the importance of native resolution in enhancing video understanding capabilities.

## C.3 Verification of the effectiveness of data scale.

From an intuitive perspective, data scale has a significant impact on the effectiveness of contrastive learning. In this section, we conduct cold-start experiments on UniViTAR-0.3B to confirm this view. For the experiment setup, seen samples is fixed at 1B. We respectively train the UniViTAR-0.3B for 1 epoch using Merged-1B and for 10 epochs using Merged-100M, which contains 100M image-text pairs that randomly sampled from Merged-1B. Result on zero-shot classification and retrieval is

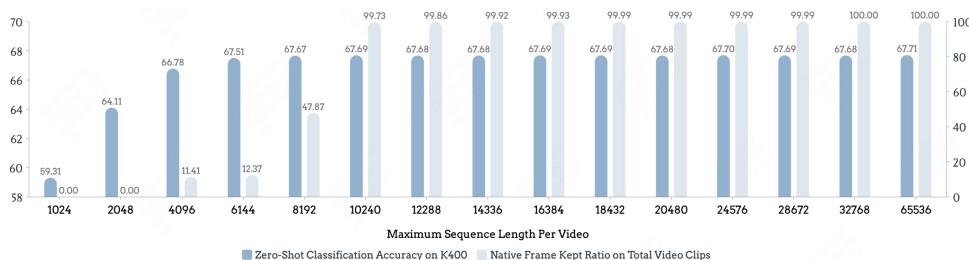

Figure 6: **Performance changes on the K400 dataset across varying sequence length limits.**

shown in Table 13. There is an observable trend where the performance improves as the dataset scale increases. With larger dataset scale, the model is exposed to a broader range of image-text pairs, facilitating a more comprehensive learning and understanding of the visual and linguistic space, thereby enhancing zero-shot performance.

Table 13: **Ablation results of UniViTAR-0.3B under varying data scale.**

| Data | Seen Samples | Overall | ImageNet Variants | | | | | | Overall | Flickr | | COCO | |
| | | | IN-1K | IN-A | IN-R | IN-V2 | IN-S | O-Net | | T→I | I→T | T→I | I→T |
|---|---|---|---|---|---|---|---|---|---|---|---|---|---|
| Merged-100M | 1B | 60.8 | 69.7 | 39.6 | 79.0 | 61.6 | 54.8 | 60.2 | 67.0 | 73.4 | 88.3 | 44.2 | 62.0 |
| Merged-1B | 1B | 64.2 | 71.7 | 45.7 | 82.3 | 64.3 | 57.3 | 63.9 | 68.9 | 74.9 | 90.7 | 46.0 | 63.8 |

## C.4 Verification of the effectiveness of LLM scale.

To verify the model's effectiveness across language models of varying scales, we integrated UniViTAR-0.6B with progressively larger language backbones, specifically, Qwen2.5-1.5B, 3B, and 7B, within a multimodal large language model (MLLM) framework. As the model scale increases, our models achieve average scores of 48.7, 51.9, and 54.6, respectively, across 16 multimodal benchmarks. The experimental results demonstrate consistent scaling behavior, confirming UniViTAR's strong compatibility and performance potential when combined with larger language models. These findings support the conclusion that the UniViTAR architecture possesses promising scalability.

## C.5 Comparative analyses of additional relevant visual foundation models.

To provide a comprehensive comparison with other prominent visual encoders, we conduct a systematic analysis of several relevant visual foundation models. (1) NaViT [3]: Quantitative evaluations in Table 14 on linear-probing classification tasks demonstrate that UniViTAR-0.3B exhibits clear advantages over NaViT. (2) FlexiViT [2]: Introduced in the main text, FlexiViT supports dynamic patch size to handle variable-resolution inputs. We include supplementary linear-probing results on ImageNet variants in Table 14 for direct comparison. (3) Web-DINO [84]: This self-supervised model shows that scaling data and parameters can approach CLIP-level performance; however, a noticeable gap remains relative to CLIP-based paradigms, as indicated by model size (7B vs. 0.3B) and benchmark performance. Preliminary comparisons are provided in the accompanying Table 14. (4) Cambrian-1 [85]: This vision-centric MLLM family uses a Mixture-of-Features (MoF) scheme over multiple visual encoders to reduce information loss. While effective, MoF introduces higher computational costs and integration complexity compared to unified models like UniViTAR.

Table 14: **Comparison of UniViTAR and other vision encoder on linear-probing classification.**

| Model | Pretrain Data | Train Paradigm | IN-1K | IN-A | IN-Real | IN-V2 | IN-S | IN-R |
|---|---|---|---|---|---|---|---|---|
| NaViT-L | JFT4B | Supervised Learning | 76.0 | 65.5 | - | - | - | - |
| FlexiViT-L | ImageNet-1K | Supervised Learning | 86.1 | 34.1 | 90.0 | 76.7 | - | 41.2 |
| Web-DINO-7B | MC-2B | Self-Supervised Learning | 86.4 | - | - | - | - | - |
| UniViTAR-0.3B | Merged-1B | Contrastive Learning | 87.6 | 84.1 | 90.3 | 79.5 | 66.0 | 90.6 |

# D Related Work

## D.1 Flexible Vision Transformers

Vision Transformers have showcased impressive performance in numerous visual tasks, such as image classification [1], image language pre-training [11], etc. Those methods work only at a single, fixed resolution. Some works [86, 87] attempt to meet the need for fine-grained visual representation by adapting the model to a higher resolution during the fine-tuning stage. However, directly resizing the input to a fixed square resolution still limits their representation capacity in diverse visual scenarios. Recently, there are some works in vision transformers attempting to accommodate images with native resolutions with variable aspect ratios. ViTAR [88] proposes an adaptive token merger module to alleviate the constraints of fixed resolution and adapt to multi-resolution inputs. However, it still limited by a predefied number of tokens that the model ultimately aims to obtain. NaViT [3] introduces sample packing used in language modeling for handling variable sequence length of image patches. Meanwhile, it introduces a factorized positional embedding schema in vanilla ViT to support variable aspect ratios and extrapolate to unseen resolutions. Qwen2.5-VL [5] integrates an NaViT-like approach to support native input resolutions, and employs multiple training phases for adapting it to multimodal large languages models, including CLIP pre-training, vision-language alignment, etc.

## D.2 Vision Foundation Models

The development of vision foundation models has progressed through distinct phases, beginning with supervised learning paradigm dominated by landmark architectures like ResNet [89] and ViT [1], which established performance benchmarks through reliance on labeled data. However, the field witnessed a paradigm shift with the rise of self-supervised learning, which circumvented annotation bottlenecks through three principal branches: contrastive learning frameworks like SimCLR [90] and MoCo [91], masked image modeling methods such as BEiT [92] and MAE [93], and self-distillation techniques including BYOL [94] and DINO [95]. Recently, language-supervised contrastive pre-training has emerged as a transformative paradigm, exemplified by CLIP [96], which aligns multimodal embeddings through noise-robust contrastive objectives, enabling zero-shot task generalization. This approach has been further refined in works like SigLIP [6], which employs a more efficient sigmoid-base loss function while preserving cross-modal transfer capabilities. Besides images, a robust visual foundation model with effective video alignment capabilities serves as another critical building block. The existing strategies for training such models can be classified into three main paradigms: training on video-only data [56, 58, 15, 97], utilizing multimodal data encompassing both video and image [15, 87, 4, 5], and incorporating multimodal data that integrate video, images, audio and other modalities [16, 98]. VideoPrism [58] employs a two-stage video-only pretraining strategy: contrastive learning followed by token distillation, yet lacks image understanding. VidLA [56] adapts CLIP [96] via spatio-temporal attention on video-text data. InternVideo [15] combines masked video modeling with alternating video/image-text pretraining, enhanced by cross-modal attention, while InternVideo2 [16] extends this framework with audio/speech modalities for multimodal alignment.

## D.3 Multimodal Large Language Models

Recently, multimodal large language models (MLLMs) have witnessed significant advancements and rapid development [99, 17, 100, 87, 5, 4, 12, 101, 102]. As a critical modality in MLLMs, visual input encounters inherent limitations when relying on conventional ViT with fixed resolutions, which may induce shape distortions, content blurring, and suboptimal handling of images/videos with diverse aspect ratios, high resolutions, or dynamic frame rates. To mitigate these challenges, the field has converged on two principal technical directions: 1) The tiling-based paradigm, as adopted by models like [103, 100, 12, 104], decomposes ultra-high-resolution inputs into a varied number of fix-resolution tiles, and each tile is processed by a fixed-resolution vision encoder. As such, it enables MLLMs adaptivity to dynamic-resolution images without padding or shape-distorting resizing. However, the tile limits the model's ability to capture spatial information across different tiles and the primary subjects of the images are often fragmented, leading to the loss of spatial relationships and quantitative information. 2) native-resolution methodology, exemplified by models such as [4, 105, 106], attempts to circumvent the limitations of the tiling-based paradigm by using native resolution input. However, they typically employ a pretrained fixed-resolution vision transformer as vision encoder, which leads to additional costs associated with adapting the ViT's distribution.

