# OpenReview forum: "VITRIX-UniViTAR: Unified Vision Transformer with Native Resolution"
_NeurIPS.cc/2025/Conference — NeurIPS 2025 poster_

### Official Review · Reviewer_xyrn · 2025-06-30

**Clarity:** 2
**Significance:** 4
**Originality:** 3
**Rating:** 4
**Confidence:** 5

**Summary:**

The paper proposes UniViTAR (Unified Vision Transformer with Native Resolution), a family of vision foundation models designed to handle visual inputs in their native resolution and aspect ratio across both image and video modalities. Unlike conventional Vision Transformers that operate on fixed-size, square inputs, UniViTAR aims to preserve more spatial and contextual fidelity by avoiding aggressive resizing.

To achieve this, the authors make several contributions:
- Versatile Framework: They introduce a family of homogeneous visual foundation models that support native resolution and unified feature extraction across visual modalities.
- Progressive Training Paradigm: A four-stage training process is introduced, beginning with fixed-resolution pretraining and gradually adapting to native resolution and video inputs.
- Empirical Performance: Trained only on public datasets, UniViTAR models (ranging from 0.3B to 1.4B parameters) achieve leading performance on a range of image and video classification/retrieval tasks, as well as strong results on multimodal benchmarks.

**Questions:**

1. Does the training scheme based on native resolution present efficiency issues? In this paper, images are resized during training to ensure that the final sequence length matches the specified maximum length. Compared to directly maintaining the image’s native resolution, this approach significantly increases computational resource requirements during training. Would a method similar to the Qwen series, which directly uses the native resolution of images for training, be a better solution?

2. Does scaling images to the maximum sequence length during training, while keeping their native resolution during inference, impact inference performance? During inference, the model may encounter very small images, whereas most images during training are enlarged to the maximum sequence length. This leads to a distribution mismatch between training and inference data, which could negatively affect the performance on small images. The poor performance of the Native Resolution approach at a sequence length of 256, as shown in the right graph of Figure 3, precisely illustrates this issue. Since many benchmarks in Table 6 include a large number of small images, it would be valuable to investigate whether appropriately increasing the resolution of these small images can further improve performance. Such an ablation study should be included.

3. How efficient is inference with ViT using native resolution? When the image resolution is very high, a large number of tokens will inevitably be generated. Even though this may lead to improved performance compared to traditional methods, will the inference efficiency decrease significantly as a result?

4. While the model is capable of handling higher-resolution inputs, it was not sufficiently trained across the full range of such resolutions. Did the authors analyze how the distribution of image resolutions and aspect ratios in the training data affects the model’s performance on tasks requiring varying input sizes?

**Ethical Concerns:**

["NO or VERY MINOR ethics concerns only"]

**Final Justification:**

The author addresses most of my concerns. I would like to keep my rating of borderline accept.

**Limitations:**

The authors did not discuss the limitations of the paper in a dedicated section. In my opinion, the main limitations of this work are as follows:

1. Lack of theoretical analysis or intuitive case. For example, under what specific circumstances does the native resolution ViT outperform traditional ViT models?

2. Lack of fair comparisons. One of the main applications of ViT is as a visual encoder in MLLMs. However, Table 6 does not successfully demonstrate the superiority of native resolution ViT in this regard. This is my primary concern.

3. Computational efficiency issues. For large images, using native resolution produces a significant number of tokens. It is not desirable to sacrifice a great deal of speed for only a marginal improvement in accuracy.

Overall, this is a meaningful work, but I think there are still some pain points that need to be addressed.I will consider raising my score if the authors can adequately address the question I raised.

**Quality:**

3

**Strengths And Weaknesses:**

Strengths
1. Originality: Currently, there is no complete training scheme for native resolution-based Vision Transformers (ViTs) available in the open-source community.

2. Significance: Native resolution-based ViTs play a crucial role in visual tasks, particularly in applications that are highly sensitive to resolution requirements, like extreme resolution and ratio.

Weaknesses

1. The comparisons lack fairness. The paper states that models are kept at native resolution during inference; however, in Table 6, benchmarks such as DocVQA, InfoVQA, and HRBench4K are known to contain images with extremely high resolutions. Using native resolution results in a much larger number of tokens (several thousand) compared to algorithms based on fixed resolutions (a few hundred tokens), making the comparisons clearly unfair. It would be more appropriate to follow the approach of LLaVA-NeXT—by using tiled sub-images to increase the token count and enhance dynamic resolution awareness for the baseline methods, or by comparing with models that have more parameters but similar computational resource usage to UniViTAR. Only if the superior performance is maintained under these fairer conditions can the effectiveness of the proposed method be truly demonstrated.

2. The paper lacks intuitive case studies and theoretical analysis. Most of the content is devoted to experimental results showing that the native resolution approach outperforms the fixed resolution approach, but it does not analyze the underlying reasons. Does the native resolution method perform better in all scenarios, or only in cases of extreme aspect ratios or resolutions? It is necessary to further analyze the advantages of native resolution on specific cases or benchmarks.

---

> ### Author Rebuttal · Authors · 2025-07-31
>
> > **Weakness 1. The comparisons lack fairness. Only if the superior performance is maintained under these fairer conditions can the effectiveness of the proposed method be truly demonstrated.**
>
> To address the reviewer’s concerns about comparative fairness, we designed controlled experiments by constraining UniViTAR’s ​**​per-image token range​**​ to align with the computational budgets of baseline encoders (SigLIP-SO400m: 729 tokens; AIMv2-H: 1024 tokens). Specifically, we enforced a token range of ​​288-1170​​ (mean≈729) and ​288-1760​​ (mean≈1,024) during training/inference. Results on three high-resolution benchmarks in below table demonstrate UniViTAR’s consistent superiority under matched computational conditions:
>
> |Model (@Qwen2.5-1.5B)|Resolution| Token Range| InfoVQA|DocVQA|HRBench4K|
> |:---:|:---:|:---:|:---:|:---:|:---:|
> |SigLIP-SO400m|378|729|23.2|32.0|45.0|
> |AIMv2-H | 448 |1024|25.8|36.2|44.5|
> |UniViTAR-0.6B |Native| 288\~1170 (729±441) |  24.6|39.1|46.0|
> |UniViTAR-0.6B |Native| 288\~1760 (1024±736)  | 27.5|45.2|46.0|
>
> > **Weakness 2. The paper lacks intuitive case studies and theoretical analysis. Does the native resolution method perform better in all scenarios, or only in cases of extreme aspect ratios or resolutions?**
>
> We sincerely appreciate the reviewer's insightful comment regarding the need for deeper theoretical analysis and intuitive case studies.
> * Theoretical advantages of native resolution processing​​.
>   * ​**​Preservation of Geometric Integrity**.​ Fixed-resolution methods inevitably introduce geometric distortion, either through peripheral information loss (center cropping) or object deformation (aspect ratio warping). In contrast, native resolution processing inherently eliminates such distortions by: (1) maintaining the original aspect ratio without spatial modification, and (2) preserving fine-grained contextual relationships across the entire image canvas.
>   * ​**​Alignment with Natural Image Resolution Distribution​**.​ Native resolution processing better reflects the inherent variability in resolution found in real-world images.
> * The native-resolution approach demonstrates superior performance or greater flexibility in most scenarios. However, it is not universally optimal. Below, we provide some intuitive case studies to delineate specific scenarios:
>   * ​**​Hardware Compatibility Constraints​**​. Modern GPU tensor cores optimize for fixed-dimension computations. Variable-length token sequences cause suboptimal hardware utilization.
>   * ​**​Extreme Aspect Ratio Scenarios (>1:100)​**​. For extreme OOD-resolution inputs, both fixed and native-resolution methods empirically fail to extract good visual features due to inherent limitations in training data coverage.
>
> > **Question 1. Does the training scheme based on native resolution present efficiency issues? Compared to directly maintaining the image’s native resolution, this approach significantly increases computational resource requirements during training.**
>
> We appreciate the reviewer's insightful questions regarding the computational efficiency of our native-resolution training scheme and the comparison to approaches like Qwen2-VL. We address these points below:
> - We contend that directly training on native resolution (analogous to Qwen2-VL) presents several disadvantages, making it a suboptimal solution:
>   * ​**​Requirement for pre-packing​**.​ To maximize data utilization efficiency and training throughput, a ​**​greedy pre-packing strategy​**​ is typically adopted (as utilized in NaViT). Critically, different packing strategies can ​compromise data randomness​​ during training.
>   * ​**​Constrained resolution distribution​**.​ The ​​resolution distribution​ becomes rigidly determined solely by the training dataset, which leads to a ​​reduction in the diversity of resolutions​  encountered during training. Our proposed strategy dynamically adjusts the token sequence length per image ​within each batch​​, based on ​​both​​ the distribution of image resolutions within the current batch and the predefined ​maximum token limit​. Consequently,  the same image may be resized to different resolutions across distinct training batches, and this inherent variability ​​ensures a richer distribution of resolutions.
> - We argue that our proposed strategy ​**​does not impose a significant increase in computational resource demands​**​. The primary reason is as follows. During training with batch size greater than 1, training frameworks invariably ​​pad variable-length visual sequences within a batch to a maximum sequence length​​ for computational efficiency and minimize ​pipeline bubble. Crucially, regardless of *Batch-Padding-to-Max-Length* strategy in Qwen2-VL or *Batch-Resizing-to-Max-Length* strategy in UniViTAR, the ​upper bound​ on computational resource requirements is ​more dictated by the configured maximum sequence length. In fact, by resizing images within each batch according to our strategy, our method actually **​​reduces the overall padding ratio​** across the training process and ​enhances computational resource utilization efficiency​​.
>
> > **Question 2. Does scaling images to the maximum sequence length during training, while keeping their native resolution during inference, impact inference performance?**
>
> - We first wish that we have correctly interpreted the reviewer’s inquiry. The proposed native-resolution training strategy does not uniformly scale images to a maximum sequence length. Instead, ​**​the resolution of each image is dynamically adjusted based on the resolution distribution of all images within its batch​**​. Under this paradigm, the same image may be resized to different resolution when processed in distinct batches, which may be ​**​upscaled​**​, ​**​downscaled​**​, or ​**​remain unchanged​**​ depending on the resolutions of other images in the same batch.
> - Regarding the visualization results for Figure 3 (right panel), under strict visual sequence length constraints (256 tokens), we think the suboptimal performance of *Native Aspect Ratio* primarily stems from UniViTAR’s early training stages bias (Stage-1/2), where a fixed sequence length of 256 tokens was employed for pretraining. Crucially, as illustrated in the Figure 3, when maximum visual sequence lengths exceed 256 tokens (e.g., ≥576), the following relationship consistently holds: Native Aspect Ratio​​ > Fix-size.
> - Exploring the impact of minimum resolution on benchmark performance is indeed a valuable suggestion. We adjust the min-token number during evaluation, and the results are shown in the table below. Interestingly, performance improvements on certain benchmarks are not linear, which aligns with the findings in Qwen2-VL. Benchmarks like OCRBench contain many very small images, and excessive enlargement of these images leads to distortion of images. Overall, the average performance across the 16 benchmarks shows little difference, indicating robustness of our model.
> | Min Tokens | Avg. Performance|HallusionBench|OCRBench|
> |:---:|:---:|:---:|:---:|
> |72 |49.3|57.4|37.0|
> |144|49.5|57.4|36.8|
> |288|49.5|57.8|36.9|
> |576|49.5|57.6|36.7|
>
> > **Question 3. How efficient is inference with ViT using native resolution? When the image resolution is very high, a large number of tokens will inevitably be generated.**
>
> The integration of native-resolution processing in ViT inherently establishes a fundamental efficiency-performance trade-off, governed by the quadratic computational complexity of self-attention mechanisms relative to token count. This implies that inference efficiency degrades progressively as token volume increases—a theoretical limitation scaling polynomially with sequence length. To mitigate computational explosion during dynamic-resolution training, we impose constraints on ​**​maximum tokens per image​**​, *i.e*., CLIP pretraining phase​ is 16,384 tokens and MLLM alignment phase​ is 5,832 tokens (after pixel-unshuffle is 2916). This calibrated token ceiling balances inference efficiency against representational fidelity, ensuring tractable resource utilization while preserving performance gains.
>
> > **Question 4. Did the authors analyze how the distribution of image resolutions in the training data affects the model’s performance on tasks requiring varying input sizes?**
>
> Our analysis reveals that native-resolution vision encoder exhibits non-trivial ​**​extrapolation capabilities​**​ to resolutions beyond its training distribution.  To quantitatively assess this, we conducted an experiment using the ​*UniViTAR-0.6B + Qwen2.5VL-1.5B*​ as baseline​. During training, the maximum token count was limited to ​2,916 tokens​​ (corresponding to visual tokens number fed into the LLM). During inference, we progressively increased the evaluation maximum token limit per image to ​5,832 (2×)​​ and ​8,748 (3×)​​ tokens. Performance across our full evaluation suite and specialized high-resolution benchmarks is summarized below:
>
> |Max Tokens (Train)|Max Tokens (Evaluation)| Avg. Performance|HRBench4K|TextVQA_VAL |
> |:---:|:---:|:---:|:---:|:---:|
> |2916|2916|49.5|46.1|50.7|
> |2916|5832 (2x)|49.6|47.3|51.4|
> |2916|8748 (3x)|49.6|48.5|51.4|
>
> Increasing the maximum token limit to 5,832 (2×) and 8,748 (3×) yields ​​consistent improvements​ on benchmarks containing substantial high-resolution content, namely ​*HRBench4K*​ and ​*TextVQA\_VAL*. Furthermore, this result complements the **low-resolution extrapolation capability** demonstrated in the Question 2, ​​collectively confirming​ the model's inherent robustness across diverse resolutions beyond its specific training distribution.

---

### Official Review · Reviewer_EmmT · 2025-07-01

**Clarity:** 3
**Significance:** 2
**Originality:** 2
**Rating:** 4
**Confidence:** 4

**Summary:**

This paper proposes UniViTAR, a family of Vision Transformer models designed to process images and videos at their native resolution and aspect ratio, addressing limitations of conventional fixed-resolution ViTs.
The architecture integrates modern components such as 2D RoPE, SwiGLU, RMSNorm, and QK-Norm, and employs a progressive training strategy that transitions from fixed to native resolution and from image to video modalities.
A hybrid loss combining sigmoid-based contrastive learning and feature distillation accelerates training.
Trained solely on publicly available image-caption data, UniViTAR achieves strong performance across a range of visual and multimodal benchmarks.

**Questions:**

None

**Ethical Concerns:**

["NO or VERY MINOR ethics concerns only"]

**Final Justification:**

my concerns are well addressed.

**Limitations:**

No
The paper does not discuss its own limitations, although mentioned in check list. For a model targeting native resolution and multimodal understanding, it would be valuable to acknowledge potential weaknesses such as computational cost, scalability, or video performance gaps.

**Quality:**

3

**Strengths And Weaknesses:**

**Strengths**

1.This paper proposes a native resolution Vision Transformer that enables effective feature extraction from images of varying resolutions. It adopts a simple yet effective progressive training strategy, and demonstrates strong performance across image classification, video understanding, and multimodal tasks.

2. This paper is well-written and easy-to-follow.

**Weakness**

1.One limitation of the paper is the lack of discussion on related work that tackles similar problems using different architectures. For example, causal convolutions, which are widely used in video generation, have also been applied in encoding tasks for both images and videos, and naturally support variable-length or multi-resolution inputs. Similarly, diffusion transformer (DiT) models for image and video generation share many architectural elements with UniViTAR, such as Transformer backbones with RoPE, QK-Norm, and progressive resolution handling. While these models target generative tasks, they address similar structural and resolution challenges.

2. While positioning UniViTAR as a vision encoder for MLLMs is arguably the most impactful application of this work, the comparison is limited and outdated. The paper mainly contrasts with earlier vision encoders, but omits recent models that specifically support native resolution, such as NaViT and FlexiViT. Furthermore, several recent works focused on multimodal understanding—like Scaling Language-Free Visual Representation Learning and Cambrian-1—propose advanced vision encoders tailored for MLLMs.

---

> ### Author Rebuttal · Authors · 2025-07-31
>
> > **Weakness 1. One limitation of the paper is the lack of discussion on related work that tackles similar problems using different architectures. For example, causal convolutions, which are widely used in video generation, have also been applied in encoding tasks for both images and videos, and naturally support variable-length or multi-resolution inputs. Similarly, diffusion transformer (DiT) models for image and video generation share many architectural elements with UniViTAR, such as Transformer backbones with RoPE, QK-Norm, and progressive resolution handling. While these models target generative tasks, they address similar structural and resolution challenges.**
>
> We sincerely thank the reviewer for this insightful and valuable comment regarding the scope of the related work discussion. Expanding our discussion to encompass these approaches would have significantly strengthened the context of our work.
>
> - ​**​Discussion of Causal Convolutions​**.​ We agree that architectures employing ​causal convolution​ represent an important alternative strategy for handling sequential data, including images and videos, particularly due to their inherent support for ​variable-length inputs​​ and multi-resolution representations. While our work primarily focuses on the quality of visual semantic-level representations and their generalization capabilities in multimodal understanding scenarios, adopting the alternative approach of causal convolutions would introduce multiple substantial limitations:
>
>   - **Loss of Global Context Modeling**. Bidirectional attention in UniViTAR (*transformer-centric* paradigm) dynamically computes relationships between all  visual tokens, enabling holistic understanding of scene context and object interactions. Causal convolutions' locality fundamentally restricts context aggregation to receptive fields defined by kernel size/depth.
>   - **Suboptimal Static Image Processing**. For non-sequential visual inputs (static images), causal convolutions impose an artificial, task-irrelevant directional bias that has no perceptual justification (​*e.g*.​, prioritizing left patches over right). This conflicts with the isotropy of natural images.
>   - **Representational Capacity Constraints**. While deep stacks of dilated convolutions can increase receptive fields, they remain fundamentally less expressive than attention mechanisms for capturing complex, long-range token dependencies. This may necessitate significantly deeper architectures to compensate, diminishing potential efficiency gains.
> - ​**​Discussion of Diffusion Transformers (DiTs):​** ​While DiTs and UniViTAR share some  architectural elements, we note DiT's primary application domain is ​*generative modeling*​, whereas UniViTAR is primarily designed for ​*discriminative representation learning*​. This key distinction in objective leads to substantial differences in training regimes, loss functions, and final application. Moreover, it is crucial to emphasize that the core contribution of UniViTAR lies in its complete training scheme for native resolution-based Vision Transformers in the open-source community and a family of homogeneous visual foundation models for multimodal undersanding research.
>
> > **Weakness 2. While positioning UniViTAR as a vision encoder for MLLMs is arguably the most impactful application of this work, the comparison is limited and outdated. The paper mainly contrasts with earlier vision encoders, but omits recent models that specifically support native resolution, such as NaViT and FlexiViT. Furthermore, several recent works focused on multimodal understanding—like Scaling Language-Free Visual Representation Learning and Cambrian-1—propose advanced vision encoders tailored for MLLMs.**
>
> We sincerely appreciate the reviewer’s valuable feedback. Below, we provide comparative analyses of additional relevant visual foundation models referenced in your comments.
>
> - **NaViT​**[1]. By introducing a token-packing strategy, NaViT pioneered the application of native-resolution training schemes in conventional Vision Transformers, concurrently exploring efficiency improvements in native-resolution processing. Detailed distinctions and advantages of UniViTAR relative to NaViT are comprehensively addressed in our response to **Reviewer dNd7**. To minimize redundancy, we refrain from reiterating these points here and kindly refer the reviewer to that discussion. For quantitative validation, we present performance comparisons between UniViTAR and NaViT on linear-probing classification, demonstrating UniViTAR-0.3B’s significant advantages.
>
> |Model (@Linear-Prob)|Pretrain Data|Train Paradigm|IN-1K|IN-A|IN-Real|IN-V2|IN-S|IN-R|
> |:----:|:-----:|:----:|:---:|:---:|:---:|:----:|:----:|:----:|
> |NaViT-L| JFT4B | Supervised Learning|76.0|65.5|-|-|-|-|
> |UniViTAR-0.3B| Merged-1B |Contrastive Learning|**87.6**|**84.1**|90.3|79.5|66.0|90.6|
>
> - **​FlexiViT​**​[2]. This work is referenced in the introduction section of our main text. FlexiViT dynamically adjusts patch sizes to generate variable-length visual token sequences, accommodating both fixed and native-resolution inputs. We acknowledge its exclusion from our primary comparative analysis and clarify two key reasons:
>
>   - ​**​Model Scale​**​: FlexiViT’s published results predominantly utilize ViT-B architectures, while UniViTAR’s smallest variant (UniViTAR-L) operates at approximately 300 million parameters, rendering direct performance comparisons inequitable.
>   - **​Training Paradigm​**​: FlexiViT-L employs supervised pretraining on ImageNet, whereas UniViTAR adopts large-scale contrastive pretraining without ImageNet data. This fundamental methodological discrepancy precludes rigorous equivalence in comparative evaluation.
>   - Nevertheless, we still provide supplemental comparisons in the following table: one is the comparison results of linear-probing on ImageNet variants (Table 1), and the other is the results of the model on the zero-shot classification and cross-modal retrieval when FlexiCLIP-B at patch-size 8 (Table 2).
>
> |Model (@Linear-Prob)|Pretrain Data|Train Paradigm|IN-1K | IN-A |IN-Real |IN-V2 |IN-S |IN-R|
> |:----:|:----:|:----:|:----:|:----:|:----:|:----:|:----:|:----:|
> | FlexiViT-L| ImageNet-1K| Supervised Learning|86.1|34.1|90.0|76.7|-|41.2|
> | UniViTAR-0.3B| Merged-1B |Contrastive Learning|**87.6**|**84.1**|**90.3**|**79.5**|**66.0**|**90.6**|
>
> |Model (@Zero-Shot)| IN-1K |Flickr30K T2I |Flickr30K I2T |COCO T2I |COCO I2T |
> |:----:|:----:|:----:|:----:|:----:|:----:|
> | FlexiCLIP-B/8|60|57|75|30|49|
> | UniViTAR-0.3B  |**81**|**84**|**95**|**55**|**71**|
>
> * **Web-SSL​**[3]. Web-DINO explores visual self-supervised learning paradigms, demonstrating that scaling both data volume and model size beyond the billion-scale can yield performance approaching that of CLIP-trained vision foundation models. However, a non-trivial performance gap persists between Web-DINO and CLIP-paradigm representations, as evidenced by model parameter disparities (7B vs. 1.4B) and benchmark comparisons. We provide preliminary performance comparisons in the following Table.
>
> |  Model   | Pretrain Data| Train Paradigm|IN-1K (Linear-Prob) | ADE20K (linear) |ADE20K (multi-scale) |
> |:----:|:----:|:----:|:----:|:----:|:----:|
> |Web-DINO ViT-7B | MC-2B |Self-Supervised Learning|86.4|42.6|52.8|
> |UniViTAR-0.3B| Merged-1B |Contrastive Learning|87.6|40.7|54.6|
> |UniViTAR-1.4B| Merged-1B |Contrastive Learning|**88.9**|**45.4**|**56.2**|
>
>   - Regarding multimodal capabilities, we excerpt key results from *Table 3* of Web-SSL[3] into the following Table for the reviewer’s reference. Notably, Web-DINO at 7B scale underperforms SigLIP-SO400M-378 in multimodal tasks. Similar observations were documented in Cambrian-1[4]. Given the necessity for rigorous multimodal benchmarking, we will incorporate expanded comparative analyses in the revised manuscript.
>
> |Model| Resolution|General|Knowledge | OCR & Chart |Vision-Centric|
> |:----:|:----:|:----:|:----:|:----:|:----:|
> |Web-DINO ViT-7B | 378 |73.9 |47.7 |50.4 |57.7|
> |SigLIP-SO400M| 384 |**75.5** |**48.2**|**55.1** |**60.8**|
>
> - **Cambrian-1**[4].   Cambrian-1 introduces a family of multimodal LLMs designed with a vision-centric approach, which mainly employs the **MoF (Mixture-of-Feature)** paradigm for vision representation with a combination of four exist vision encoders: *OpenAI CLIP ViT-L/14@336, SigLIP ViT-SO400M/14@384, OpenCLIP ConvNeXt-XXL@1024, and DINOv2 ViT-L/14@518*. Considering that UniViTAR primarily focuses on (1) exploring large-scale pre-training strategies for a ​**​single visual encoder​**​ under ​native-resolution setting​ and (2) improving the quality of ​**​general visual representations​**​, we have provided detailed comparisons in the paper between UniViTAR and the CLIP series models employed in Cambrian-1 (see Experiments 3 to 6 in our paper). Generally, the MoF fusion scheme reduces information loss by integrating features from multiple visual representations. However, this approach typically incurs ​​increased computational overhead​​ and introduces complexities in selecting and integrating diverse ​​visual foundation models​​ for practical applications.
>
> To ensure ​​comprehensive comparison​, we will incorporate detailed discussions of these points in the ​updated version​​ of our manuscript.
>
> *[1] Dehghani, Mostafa, et al. "Patch n’pack: Navit, a vision transformer for any aspect ratio and resolution." *Advances in Neural Information Processing Systems* 36 (2023): 2252-2274.*
>
> *[2] Beyer, Lucas, et al. "Flexivit: One model for all patch sizes." ​*Proceedings of the IEEE/CVF Conference on Computer Vision and Pattern Recognition*​. 2023.*
>
> *[3] Fan, David, et al. "Scaling language-free visual representation learning." *arXiv preprint arXiv:2504.01017* (2025).*
>
> *[4] Tong, Peter, et al. "Cambrian-1: A fully open, vision-centric exploration of multimodal llms." *Advances in Neural Information Processing Systems* 37 (2024): 87310-87356.*

---

> > ### Author Response · Authors · 2025-08-06
> >
> > Dear Reviewer,​​
> >
> > ​​We sincerely appreciate the time and effort you dedicated to reviewing our manuscript and providing your valuable suggestions.​​ ​​ As the rebuttal discussion period concludes in two days, **we are writing to kindly inquire whether our responses have adequately addressed the concerns you raised**.  We would be immensely grateful for any feedback you may share at this stage.​​ ​​ Should you require further clarification or have any additional questions regarding our rebuttal, **please do not hesitate to let us know**.  We remain available to provide any necessary information and are happy to continue this discussion. Thank you again for your time and commitment.​​
> >
> > ​​Best regards,​​
> >
> > ​​The Authors​

---

> > ### Comment · Reviewer_EmmT · 2025-08-07
> > **response**
> >
> > Thanks authors for their detailed feedback. My concerns are well addressed. Hence I will raise my score to borderline acc.

---

### Official Review · Reviewer_G79o · 2025-07-01

**Clarity:** 3
**Significance:** 3
**Originality:** 3
**Rating:** 4
**Confidence:** 4

**Summary:**

This paper proposes UniViTAR, a family of unified Vision Transformer backbones that support native resolution and both image/video modalities, aiming to serve as a solid visual encoder for multimodal large models. The key contribution is not in architectural novelty, but in the systematic integration of community-accepted best practices—such as 2D Rotary Position Embedding, dynamic patch tokenization, RMSNorm, QK-Norm, and progressive training strategies including staged curriculum from fixed to native resolution, as well as inter-batch image/video alternation. The authors conduct extensive and rigorous experiments on public datasets, covering zero-shot classification, retrieval, linear probing, and dense prediction tasks, consistently demonstrating strong performance scaling across model sizes.

**Questions:**

1. What concrete differences (in architecture, training, or results) set UniViTAR apart from Qwen2VL's vision encoder? A table would help clarify any unique contributions.

2. How much benefit can be gained by using Dinov2 for distillation in Stage 1?

3. On vision-language benchmarks, the performance gain of scaling the vision encoder from 0.6B to 1B seems to be limited. What is the author's perspective on this phenomenon?

**Ethical Concerns:**

["NO or VERY MINOR ethics concerns only"]

**Final Justification:**

The author’s response addressed my questions well. I think this is a fairly solid piece of work, so I will keep my score of 4.

**Limitations:**

See the weaknesses.

**Quality:**

3

**Strengths And Weaknesses:**

# Strengths

1. The implementation is solid, with carefully staged training, systematic ablation studies, and thorough benchmarking across diverse visual tasks (images, video, multimodal VQA, dense prediction, etc.).

2. The methodology, training details, and experimental setups are clearly described, with extensive tables and analyses covering key baselines, scaling trends, and ablation results.

3. The work serves as a robust “best practices” summary for engineering strong vision backbones for multimodal LLMs, and the systematic experiments further validate the effectiveness of these recipes.

# Weaknesses

The core architectural design and training strategy closely follow established approaches in the community, especially Qwen2VL and NaViT. The work mainly consolidates community best practices rather than proposing new model components or learning algorithms.

---

> ### Author Rebuttal · Authors · 2025-07-30
>
> > **Weakness 1. The core architectural design and training strategy closely follow established approaches in the community, especially Qwen2VL and NaViT. The work mainly consolidates community best practices rather than proposing new model components or learning algorithms.**
>
> > **Question 1. What concrete differences (in architecture, training, or results) set UniViTAR apart from Qwen2VL's vision encoder? A table would help clarify any unique contributions.**
>
> We sincerely appreciate the reviewer’s perspective regarding UniViTAR’s consolidation of community best practices and value their constructive feedback. Pioneering works such as NaViT introduced native-resolution training schemes for Vision Transformers (ViTs), exploring efficiency improvements in native-resolution processing. Similarly, Qwen2VL’s vision encoder adopted NaViT’s principles by leveraging pretrained fixed-resolution visual foundation model (​*e.g*.​, DFN[1]) to advance multimodal representation alignment and pretraining for large multimodal models. In our response to ​**​Reviewer dNd7​**​, we comprehensively contrasted UniViTAR’s distinctions and advantages relative to NaViT and Qwen2.5-VL’s vision encoder. To avoid redundancy, we kindly invite you to refer to the relevant sections of our reply to **Reviewer dNd7** for more detailed comparative analyses. Below, we explicitly delineate key differences between the UniViTAR framework and Qwen2-VL’s vision encoder:
>
> | Model          | Qwen2-VL-ViT               | UniViTAR family                                                         |
> | -------- | -------- | ---- |
> | Architecture        | Inherit from DFN [1]      | Redesign based on the latest advanced modules                   |
> | Model size          | 0.6B                       | **0.3B, 0.6B, 1.4B**                                          |
> | Norm layer          | LayerNorm                  | RMSNorm                                                           |
> | FFN layer           | Vanilla MLP                        | SwiGLU                                                            |
> | QK norm             | No                         | Yes                                                               |
> | **Training**     |  |                       |
> | Pretrain weight     | Load from DFN          | **Scratch training**                                                  |
> | Supervised paradigm | Next text token prediction | Image-Text & Video-Text contrastive learning                                   |
> | Train data          | In-house-Data              | One billion from opensource data                                    |
> | Batch strategy      | Sequence Padding           | Sequence Resizing (See Section 2.2.2)                             |
> | Resolution strategy | Native-resolution training | Pretrained on fixed-resolution and finetuned on native-resolution |
>
>
> > **Question 2. How much benefit can be gained by using Dinov2 for distillation in Stage 1?**
>
> Thank you for your perceptive and highly professional inquiry. We did conduct a thorough investigation into the performance gains offered by the distillation branch during the initial model exploration phase. The following summarizes our empirical observations and experimental findings:
>
> -  ​**Acceleration of early-stage convergence**.​  Our experiments on UniViTAR-0.3B involved training on 3B image-text samples from scratch. Evaluating checkpoints at different training intervals (every 0.5B samples) within the same training cycle revealed that utilizing Dinov2 for distillation significantly enhances convergence during the early training phase. For instance, at 0.1B samples, distillation supervision yielded a performance gain of 17.3 points on the ImageNet-1K zero-shot classification. This beneficial effect diminishes progressively as training advances, as detailed in the subsequent comparison.
>
> |        Model       | Dinov2 | 0.1B | 0.5B | 1.0B | 1.5B | 2.0B |2.5B |3.0B |
> |:--------------------:|:---------------:|:------:|:-------:|:-------:|:-------:|:-------:|:-------:|:-------:|
> | UniViTAR-0.3B-Stage1 |    No      | 26.88|63.71|67.80|69.75|72.32|74.88|75.72|
> | UniViTAR-0.3B-Stage1 |  Yes  | 44.18|68.15|71.24|73.17|74.71|76.40|77.33|
> | *Performance Gain*        |                    | *17.30*|   *4.44*|  *3.44*|  *3.42*|  *2.39*|  *1.52*|  *1.61*|
>
> - ​**​Improvement in final model performance**.​ Evaluation of the final model trained on 3B image-text pairs demonstrated that employing Dinov2 as the distillation branch consistently delivers an average performance gain of 2.1 points across six zero-shot classification benchmarks. The detailed comparative analysis is presented below.
>
> |        Model       | Dinov2 | Seen | Avg. | IN-1K | IN-A |IN-R |IN-V2 |IN-S |O-Net |
> |:--------------------:|:---------------:|:------:|:-------:|:-------:|:-------:|:-------:|:-------:|:-------:|:-------:|
> | UniViTAR-0.3B-Stage1 |    No      | 3.0B |70.73|75.72|58.76|87.98|68.22|62.95|70.78|
> | UniViTAR-0.3B-Stage1 |  Yes     | 3.0B |**72.84**|77.33|63.55|89.81|70.40|65.65|70.31|
>
> > **Question 3. On vision-language benchmarks, the performance gain of scaling the vision encoder from 0.6B to 1B seems to be limited. What is the author's perspective on this phenomenon ?**
>
> We sincerely appreciate this insightful observation regarding the scaling behavior of vision encoders in vision-language models. As the capacity of the visual encoder increases, we observe that the VLM model's performance on established vision-language benchmarks becomes increasingly **constrained by factors other than pure visual representation power**. We would like to humbly submit the following perspectives on this phenomenon:
>
> - ​**​Cross-modal Alignment Limits**.​ Larger vision encoders inherently possess greater potential for highly specialized visual feature extraction. However, the capacity and optimization of the modality fusion mechanism may become a bottleneck, limiting the utility of richer visual features.
> - ​**Late-Fusion[2] Architecture Bound**.​  Current late-fusion methodologies typically entail integrating independently pre-trained modality-specific models, such as connecting vision encoders to LLMs and continuing multimodal training. Crucially, each constituent module (*e.g.*, vision encoder, LLM) exhibits distinct hyperparameters, distinct pre-training data mixtures, and distinct scaling properties with respect to computational resources and training data volume. Exploring the joint scaling strategy of visual encoders and LLMs may be the most necessary point in the future work.
> - ​**​Limited Train Data and Benchmark Saturation**.​ Our models in Table 6 were trained on a comparatively limited multimodal dataset, *i.e.* research-level, consisting of only 2.5M image-caption pairs and 665K instruction tuning samples. Furthermore, the visual encoder parameters remained frozen throughout training for more accurately verifing the ability of pre-trained visual representation. This configuration resulted in diminishing performance gains relative to increasing visual encoder capacity. On the other hand, some benchmarks may lack the granularity or difficulty to fully reflect the nuanced improvements gained from larger vision encoders, leading to apparent saturation.
>
>
> *[1]. Fang, Alex, et al. "Data filtering networks." *arXiv preprint arXiv:2309.17425* (2023).*
>
> *[2]. Shukor, Mustafa, et al. "Scaling laws for native multimodal models." *arXiv preprint arXiv:2504.07951* (2025).*

---

### Official Review · Reviewer_dNd7 · 2025-07-03

**Clarity:** 3
**Significance:** 3
**Originality:** 2
**Rating:** 4
**Confidence:** 3

**Summary:**

The paper introduces UniViTAR, a family of homogeneous vision foundation models designed to process native-resolution visual data (images and videos) uniformly, addressing limitations of conventional fixed-resolution Vision Transformers (ViTs). It introduces architectural enhancements (like 2D Rotary Positional Embeddings, SwiGLU, RMSNorm, and QK-Norm) and a progressive training strategy combining curriculum learning, modality alternation, and contrastive learning with feature distillation. Trained on public datasets, UniViTAR scales from 0.3B to 1.4B parameters and achieves strong performance across zero-shot image/video classification, retrieval, and multimodal understanding tasks.

**Questions:**

How does UniViTAR differentiate itself from prior native-resolution approaches like NaViT and Qwen2.5-VL?

**Ethical Concerns:**

["NO or VERY MINOR ethics concerns only"]

**Final Justification:**

I'm not totally convinced about the novelty of their native-resolution idea. That said, the paper makes meaningful engineering contributions, and I would recommend it for a borderline accept.

**Limitations:**

yes.

**Paper Formatting Concerns:**

no concerns

**Quality:**

3

**Strengths And Weaknesses:**

Strengths:
1. Open-sourcing benefits the community. Training on publicly accessible datasets and releasing models/code promotes reproducibility and encourages further research.

2. UniViTAR achieves competitive or superior results on a broad range of benchmarks across image, video, and multimodal tasks.

3. The integration of advanced components (e.g., 2D-RoPE, SwiGLU, QK-Norm) and the staged curriculum (e.g., resolution and modality progression) shows a well-engineered and scalable framework.

Weaknesses:
1. While the paper emphasizes native resolutions, existing models like Qwen2.5-VL already adopt native resolution strategies, reducing the novelty of the contribution.
2. The multimodal performance (Table 6) is relatively modest, and extensions to large-scale MLLMs remain under-explored.
3.  The paper would benefit from more granular ablation studies on the architectural choices (e.g., LayerScale, RMSNorm) and their impact across both classification and multimodal tasks.

---

> ### Author Rebuttal · Authors · 2025-07-31
>
> > **Weakness 1. While the paper emphasizes native resolutions, existing models like Qwen2.5-VL already adopt native resolution strategies, reducing the novelty of the contribution.**
>
> > **Question 1. How does UniViTAR differentiate itself from prior native-resolution approaches like NaViT and Qwen2.5-VL?**
>
> We sincerely appreciate the reviewer’s insightful perspective regarding native-resolution approaches and their valid inquiry into UniViTAR’s differentiation. We consider Qwen2.5-VL and NaViT to be significant as previous implementations of native-resolution processing. However, UniViTAR introduces ​several core innovations​​ that fundamentally differentiate it from these prior efforts in terms of both technology and functionality.
>
> **Comparison with NaViT​**​. As an early exploratory work, NaViT's core contribution lies in introducing a native-resolution token packing training strategy to Vision Transformers (ViTs), thereby pioneering efficiency improvements in native-resolution training. While its insights indeed provide foundational inspiration for contemporary native-resolution models, UniViTAR demonstrates distinct advantages:
>
> - ​**​Scalability and Homogeneity​**​. UniViTAR offers larger and more diverse model scales (0.3B/0.6B/1.4B), with all models sharing structural and training homogeneity across sizes and data sources. This facilitates granular community research into scaling both vision foundation model and multimodal large language model.
> - ​**​Flexible Native-Resolution Training Strategy​**​. NaViT primarily employs greedy token packing with sequence padding to fixed lengths during training. In contrast, UniViTAR adopts a random packing strategy combined with batch-wise sequence resizing (#L149~#L153).
> - ​**​More Visual Modality​**​. UniViTAR employs a multi-stage training paradigm (image pretraining → image-video joint training), achieving state-of-the-art performance on both image and video benchmarks. NaViT, limited to image-only pretraining, underperforms many CLIP-based counterparts.
> - ​**​Advanced Architectural Components​**​. For transformer-centric native-resolution modeling, adaptive positional encoding is essential: while NaViT uses factorized positional embeddings, UniViTAR implements 2D-RoPE to encode visual sequences with variable lengths, critically enhancing positional awareness.
>
> **Comparison with Qwen2.5-VL-ViT​**​. As a leading open-source Vision-Language Model (VLM), Qwen2.5-VL similarly adopts a native-resolution multimodal training strategy akin to NaViT. UniViTAR demonstrates the following distinctive advantages:
>
> - **​Purer Native Dynamic-Resolution Representation​**​. Qwen2.5-VL’s vision encoder undergoes extensive multimodal training (primarily on in-house data) across multiple multimodal stages (multimodal pretraining → post-training), resulting in representations highly adapted with Qwen-series LLMs. This alignment substantially constrains ​generic visual transferability​. UniViTAR, conversely, employs contrastive learning only for native-resolution pretraining, delivering superior generic visual representations.
> - **​Granular Training Recipe and Model Scalability​**​. While Qwen2.5-VL releases pretrained VLM weights, it withholds the visual pretraining weights in CLIP phase and critical implementation specifics. UniViTAR systematically documents training methodologies and releases multi-scale models (0.3B/0.6B/1.4B VS. 0.6B), establishing a comprehensive research foundation for the community.
> - ​**​Architectural Distinction in Attention Mechanism​**​. UniViTAR utilizes ​​Full-attention​ exclusively, whereas Qwen2.5-VL employs a hybrid ​​Full-attention and Window-attention​ design. Although window-attention improves computational efficiency, it typically incurs non-trivial performance degradation in visual representation quality. Considering the versatility of downstream tasks and the fact that the scale of language models is usually much larger than that of visual encoders, UniViTAR uses Full-Attention to preserve the visual representation capabilities as much as possible. However, improving the training and inference efficiency of visual encoder at native-resolution will be our next important work.
>
> > **Weakness 2. The multimodal performance (Table 6) is relatively modest, and extensions to large-scale MLLMs remain under-explored.​**
>
> We sincerely appreciate the reviewer’s insightful observation regarding the multimodal performance in Table 6 and the need for large-scale MLLM exploration. Due to the research-level multimodal traning data, we acknowledge that UniViTAR’s current multimodal results are not state-of-the-art, as our primary focus was rigorously validating the ​**​generic visual representation quality​**​ of dynamically scaled vision encoders and other state-of-the-art vision encoders—rather than optimizing end-to-end MLLM systems.
>
> To address reviewer's concren, we have conducted new experiments scaling UniViTAR to larger MLLM frameworks, e.g., integrating UniViTAR-0.6B with Qwen2.5-1.5B/3B/7B. Results demonstrate clear scaling laws in the table below, which confirms UniViTAR’s scalability potential when paired with larger language models.
>
> | ​LLM Scale​| ​Avg. on overall 16 benchmarks​ |
> | ------| :-----:|
> |UniViTAR-0.6B + Qwen2.5-1.5B| 48.7|
> |​​UniViTAR-0.6B + Qwen2.5-3B ​  | ​​51.​9​ |
> |​​UniViTAR-0.6B + Qwen2.5-7B ​  | 54.6 |
>
> Exploring UniViTAR's generalization capabilities on larger language models (​*e.g.*​, 70B parameters) and expanded data scales constitutes our critical next work to further validate the versatility of the UniViTAR family. We thank the reviewer for highlighting this gap—it directly shapes a more impactful contribution.
>
>
> > **Weakness 3. The paper would benefit from more granular ablation studies on the architectural choices (e.g., LayerScale, RMSNorm) and their impact across both classification and multimodal tasks.**
>
> We sincerely thank the reviewer for the constructive feedback regarding the need for more granular ablation studies. We acknowledge that deeper analysis of architectural components is critical for understanding UniViTAR’s design efficacy. To address this, we established a validation baseline based on UniViTAR-0.3B, with the overall training size controlled at 0.5B scale (Merge1B for 0.5 epoch). We validated the effectiveness of the three modules, LayerScale, RMSNorm, and SwiGLU. The specific results are as follows:
>
> |  Model (@Zero-Shot Classification)      | Norm Layer | FFN Layer | LayerScale | Avg. | IN-1K | IN-A |IN-R |IN-V2 |IN-S |O-Net |
> |------------|:------------:|:------:|:-------:|:-------:|:-------:|:-------:|:-------:|:-------:|:-------:|:-------:|
> | UniViTAR-0.3B (baseline) |  LayerNorm | MLP | No |58.25|68.12|33.47|78.09|59.67|53.40|56.76|
> | UniViTAR-0.3B | **RMSNorm**| MLP| No | 59.00|68.26|34.23|78.82|60.92|54.21 |57.57|
> | UniViTAR-0.3B |  LayerNorm | **SwiGLU** | No|60.00|**68.93**|36.63|79.74|60.94|55.03|**58.71**|
> | UniViTAR-0.3B | LayerNorm | MLP |**Yes**|58.58 |68.16|34.39|77.66|60.62|52.85| 57.81|
> | UniViTAR-0.3B | **RMSNorm** |**SwiGLU**|**Yes**|**60.59**|68.89|**37.37**|**81.05**|**61.70**|**56.16**|58.40|
>
> The ablation studies summarized in Table yield the following conclusions:
>
> - All three investigated modules—SwiGLU, RMSNorm, and LayerScale—enhance visual representation capabilities, albeit to varying degrees. SwiGLU yields the most substantial gains (+1.75%), followed by RMSNorm (+0.75%), and LayerScale (+0.33%). Their synergistic integration delivers cumulative performance improvements (+2.34%), confirming complementary benefits.
> - SwiGLU and RMSNorm, widely adopted in large language models (*e.g.*, LLaMA), demonstrate comparable efficacy in visual representation learning. This validates their generalizability as universal architectural components across modalities.
>
> These ablation results and associated discussions will be incorporated into the revised manuscript to clarify the impact of individual design choices.

---

### Decision · Program_Chairs · 2025-09-17

**Decision:**

Accept (poster)

**Comment:**

This paper introduces UniViTAR, a family of vision foundation models designed to process images and videos in their native resolutions. The work consolidates best practices (e.g., 2D RoPE, SwiGLU, RMSNorm, QK-Norm) into a unified training recipe, scaling models from 0.3B to 1.4B parameters and demonstrating strong performance across classification, retrieval, and multimodal benchmarks.

The main critique from reviewers was limited novelty, as the design draws heavily from NaViT, Qwen2VL, and related works. Reviewers also noted the need for fairer comparisons, more granular ablations, and broader discussion of related approaches. The authors addressed these concerns in rebuttal with new controlled experiments, detailed ablations, expanded related work discussion, and additional multimodal scaling results.

Given the rebuttals from authors, all reviewers ultimately recommended borderline accept, recognizing the paper’s solid engineering contributions, clear empirical validation, and community value through open-sourced models and training recipes. The ACs went through the discussions and final recommendations from each of the reviewers, and recommend an acceptance for this work. While the conceptual novelty is incremental, the work is technically sound, and likely to have meaningful impact as a reliable visual backbone for multimodal research. The authors should incorporate the experiments and discussions into the final revision, and release the code and model for reproduction and downstream uses.